# Entomological surveys and insecticide resistance in the dengue vector *Aedes aegypti* in Dakar, Senegal: First detection of the *kdr* mutation

Ndeye Marie Sene[1]*, Shirley Nimo-Paintsil[2], Moussa Gaye[1], El Hadj Ndiaye[1], El Hadji Malick Ngom[1], Babacar Diouf[1], Faty Amadou Sy[1], Moussa Moise Diagne[3], Alioune Gaye[1], Diawo Diallo[1], Ibrahima Dia[1], Scott C. Weaver[4], Samuel Dadzie[5], James F. Harwood[6], Mawlouth Diallo[1]

1 Medical Zoology Pole, Institut Pasteur de Dakar, Dakar, Senegal, 2 United States of America Naval Medical Research Unit-EURAFCENT Ghana Detachment, Accra, Ghana, 3 Pôle de Virologie, Institut Pasteur de Dakar, Dakar, Senegal, 4 World Reference Center for Emerging Viruses and Arboviruses, Institute for Human Infections and Immunity, Department of Microbiology and Immunology, University of Texas Medical Branch, Galveston, Texas, United States of America, 5 Noguchi Memorial Institute for Medical Research, Legon, Accra, Ghana, 6 United States of America Naval Medical Research Unit-EURAFCENT Sigonella, Italy

* ndeyemarie.sene@pasteur.sn

## Abstract

*Aedes aegypti* is the primary vector of arboviruses in Senegal, yet this species is not typically targeted by routine vector control programs. Through entomological surveillance, we investigated over a one-year (2022–2023) the risk of arbovirus transmission in Dakar, Senegal, the spatial distribution of insecticide resistance and the underlying resistance mechanisms. Weekly ovitraps were deployed in 15 localities (10 per locality), and monthly adult mosquito collections were conducted in six localities. Arboviruses were detected in adult *Ae. aegypti* using Real-time quantitative reverse transcription polymerase chain reaction (RT-qPCR) and positive sample were sequenced for phylogenetic analysis to determine the genetic diversity. Blood-feeding preferences and resting behaviors were assessed, and WHO tube bioassays evaluated susceptibility to pyrethroids, organophosphates, and carbamates. Molecular screening targeted knockdown resistance (*kdr*) mutations (F1534C, V1016G/I, S989P). Ovitrap positivity peaked between August and October, coinciding with the rainy season. *Ae. aegypti* was detected in all surveyed localities, and dengue virus (DENV) was found in adult mosquitoes between September and December 2022, confirming active transmission risk. Females predominantly fed on humans and were found resting both indoors and outdoors. All tested populations were resistant to permethrin and bendiocarb, with suspected resistance to malathion. *kdr* mutations of F1534C (CC, FC), V1016G (VG), S989P (SP, and PP) were detected. These findings demonstrate a substantial entomological risk of arbovirus transmission in Dakar, characterized by high vector density, confirmed

**Data availability statement:** All relevant data are within the manuscript and its Supporting Information files.

**Funding:** This work was supported by Armed Forces Health Surveillance Division, Global Emerging Infections Surveillance (AFHSD/GEIS) Branch; (PROMIS numbers P0041_22_N3, P0036_23_N3 to James F. Harwood) and the West African Center for Emerging Infectious Diseases (NIH/NIAID) (U01AI151801 to Scott C. Weaver). The funders had no role in study design, data collection and analysis, decision to publish, or preparation of the manuscript.

**Competing interests:** The authors have declared that no competing interests exist.

DENV infection, and widespread insecticide resistance. In this study, we detected three *kdr* mutations (F1534C, V1016G, and S989P), while previous studies in Senegal have documented metabolic resistance mechanisms. Together, these resistance mechanisms may compromise the efficacy of vector control strategies. Integrated vector management combining rational insecticide use, source reduction, and innovative control tools is recommended for sustainable *Aedes*-borne disease control in Senegal.

## Author summary

Weekly ovitrap indices and monthly adult parameters of *Ae. aegypti* were evaluated in 15 neighborhoods of Dakar, Senegal, over a one-year (2022–2023). The spatial distribution of insecticide resistance, the blood-feeding preferences, the underlying mechanisms and arboviruses infection on this vector were also investigated. The highest proportions of homes with *Aedes*-positive traps were observed between August and October. *Aedes aegypti* was found in all surveyed sites. DENV was detected in *Ae. aegypti* collected between September and December 2022. Mosquito populations from all sites exhibited resistance to the newly developed WHO diagnostic insecticide doses for *Aedes* mosquitoes. Low frequencies of the kdr mutations 1534C, V1016G, and 989P were identified. The study highlighted the risk of *Aedes*-borne disease outbreaks during the rainy season in Dakar. Multiple insecticide resistance mechanisms detected in the study populations could impact the effectiveness of control measures. Improving sanitation should be considered alongside insecticide-based strategies for *Ae. aegypti* control in the study areas.

## Introduction

*Aedes aegypti* is the primary urban vector of several arboviruses, including dengue, Zika, chikungunya, and yellow fever. In Senegal, it is responsible for all urban outbreaks in recent years, particularly in the capital Dakar [1]. Despite its major role, data on the species bioecology remain limited [2]. In recent years, dengue has emerged as a significant public health concern in Senegal. Major outbreaks occurred between 2017 and 2019 in several regions including Dakar, Touba, Fatick, Louga, and Saint-Louis, with over 420 confirmed cases [3], followed by another outbreak in Rosso in 2021, with more than 100 confirmed cases [4]. Globally, dengue is one of the most significant re-emerging infectious diseases [5]. Its etiologic agent, dengue virus (DENV), belongs to the *Orthoflavivirus* genus and comprises four distinct serotypes (DENV-1 to DENV-4). While DENV-2 circulates in a sylvatic cycle in Africa involving monkeys and *Aedes* species, all four serotypes can be transmitted in urban settings by *Ae. aegypti* [2]. The distribution and expansion of arboviruses and their

vectors are influenced by both environmental and anthropogenic factors. However, in Senegal, data on the bioecology of *Ae. aegypti* particularly its biting behavior and resting patterns remain limited.

Given the no specific treatment or commercially available vaccine for dengue and other *Ae. aegypti*-borne diseases in many regions, vector control remains the most effective tool to prevent and mitigate potential outbreaks. Developing an efficient vector control tool against this species is a major challenge for health authorities in Senegal. Currently, the emergency vector control interventions primarily involve space spraying of pyrethroids during dengue outbreaks in urban areas [6,7]. In some instances, larval source management and public awareness campaigns have also been implemented. These measures are often deployed reactively and lack long-term integration into routine control programs. However, the emergence and spread of insecticide resistance in *Ae. aegypti* could severely compromise the effectiveness of these strategies. Previous studies have reported resistance to multiple classes of insecticides, including pyrethroids, organophosphates, and carbamates. For instance, Dia et al. (2012) [6] first documented signs of resistance to permethrin in Dakar, and later Sene et al. (2021) [7], confirmed broader resistance patterns and explored underlying mechanisms in *Ae. aegypti* populations across Senegal. These findings raise significant concerns about the sustainability and efficacy of current insecticide-based control efforts. While the continued use of pyrethroids may appear suboptimal given the growing evidence of resistance, characterizing the presence and distribution of *kdr* mutations remains important for informing vector control strategies. Different *kdr* mutations can vary in their impact on resistance intensity, and combinations of mutations may further elevate resistance levels. Moreover, resistance is often heterogeneous across space and time. Therefore, spatiotemporal surveillance and *kdr* genotyping are valuable tools to guide adapted, evidence-based interventions, particularly in the context of outbreak response. Resistance mechanisms in *Aedes* mosquitoes worldwide generally involve target-site mutations and metabolic detoxification [8,9]. Several knockdown resistance (*kdr*) mutations have been identified as resistance markers in *Aedes* mosquitoes globally [10].

In Senegal, previous studies reported the total absence of *kdr* mutations (S989P, F1534C, V1016G/I), suggesting that, *Ae. aegypti* populations mainly rely on metabolic resistance mechanisms involving detoxification enzymes [7]. However, some of these mutations, such as F1534C and V1016I, are present in several populations in neighboring countries, including Burkina Faso, Côte d'Ivoire, Cape-Verde, Niger, and Mauritania [11–15]. The V1016G and S989P mutations, first described in *Ae. aegypti* populations from Asia, have been reported in several African countries, including Benin and Mauritania indicating a possible geographic spread of these resistance [13,16]. These mutations have been associated with strong resistance to pyrethroids, particularly when combined with other *kdr* mutations such as F1534C, highlighting the importance of monitoring them in African *Ae. aegypti* populations. It is possible that these mutations are also present in Senegal populations but due to limited surveillance efforts, they have yet to be detected. Furthermore, the co-occurrence and potential synergistic effects of different *kdr* mutations can elevate resistance intensity and reduce the efficacy of insecticide-based interventions. Therefore, molecular surveillance of *kdr* mutations, alongside phenotypic resistance monitoring, remains valuable for adapting vector control strategies to local resistance profiles and for optimizing the use of available insecticides. The coexistence of *kdr* mutations and metabolic resistance mechanisms, has been widely documented in *Ae. aegypti* populations across Southeast Asia, sub-Saharan Africa and Latin America [17–20]. The detection of such combinations in Senegal is therefore particularly concerning for the efficacy of current pyrethroid-based control strategies.

Vector control strategies should be guided by longitudinal bio-ecological data on *Ae. aegypti* (host-seeking, resting and feeding behaviors, infection patterns, insecticide susceptibility) in urban areas where outbreaks occur. In Senegal, such data remain limited particularly, in the capital, Dakar.

The objectives of this study were to: (i) establish baseline data on the distribution and behavior of *Ae. aegypti* populations, including egg density over space and time, host-seeking, host-preferences, and resting behavior; (ii) assess the infection rates of collected mosquitoes with arboviruses; and (iii) characterize the insecticide resistance status and resistance mechanisms in the populations across various study sites.

## Materials and methods

### Ethics statement

These studies performed through the NIH and AFHSD/GEIS projects were conducted in accordance with the guidelines established by the Senegalese Ministry of Health. This study was approved by the Comité National d'Ethique pour la Recherche en Santé (CNERS) (N°000089 and 000027/MSAS/ DPRS/CNERS), SEN20/08, approved on June 2021 and February 2023. The study protocol was carefully explained to the chief and inhabitants of each neighborhood investigated to obtain their informed oral consent. Informed oral consent was also obtained from the heads of each household in which mosquito samples were collected.

### Study area

An entomological surveillance system was implemented in Dakar (14° 45' 52.215" N 17° 21' 57.703" W). Dakar, the most populated city in Senegal with 3,896,564 inhabitants, lies at the westernmost point of continental Africa. Highly urbanized, it is characterized by wooded savannah vegetation and a climate with two distinct seasons: a hot and humid rainy season from June to October and a dry season from November to May. The mean annual temperature is approximately 27˚C, with total annual rainfall between 300- and 600-mm. Relative humidity varies between 52% in December and 75% in August.

Administratively, Dakar is divided into five districts: Dakar, Guédiawaye, Pikine, Rufisque and Keur Massar. Mosquito sampling was conducted in 15 localities spanning four of these districts: Dakar (Liberté, Point E, Grand Dakar, Ngor, Ouakam, Yoff, Grand Yoff, Médina, Plateau, Parcelles and Hanne village), Guédiawaye (Guédiawaye), Pikine (Mbao and Pikine) and Keur Massar (Keur Massar). Collections were performed during both the rainy (June to October) and the dry season (November to May), covering a full annual cycle from July 2022 to August 2023 (Fig 1).

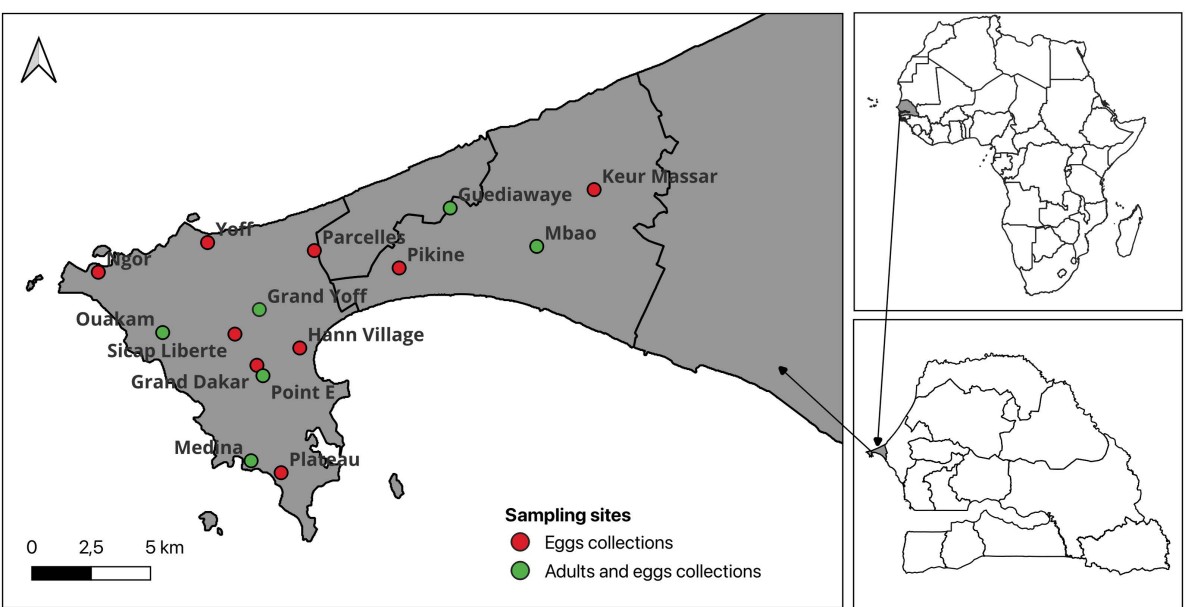

**Fig 1. Map of the city of Dakar showing the study sites.** This map was created using the R software (version 4.4.0) and the package raster using an empty shapefile from the HDX website (https://data.humdata.org/dataset/senegal-administrative-boundaries) available under Creative Commons Attribution 4.0 International licence.

## Ovitrap surveillance

Ovitrap surveys were conducted weekly following the epidemiological weeks (standardized weekly reporting periods defined by the World Health Organization, beginning with the first week of the year that includes at least four days of January) from July 2022 to August 2023 in 15 localities. In each locality, 10 houses were selected, resulting in a total of 150 ovitraps deployed. The ovitraps consisted of 0.5 L black plastic containers placed 1.5 m above the ground inside randomly selected houses, following informed consent from the homeowners. Each ovitrap was filled with water and included a wooden stick serving as an oviposition substrate. Oviposition sticks were collected and replaced weekly, and eggs were counted under a stereomicroscope.

## Adult mosquito sampling

Adult mosquitos' collections were conducted monthly from August 2022 to July 2023 in six selected localities (Médina, Ouakam, Point E, Grand Yoff, Guédiawaye and Mbao). These localities were selected based on logistical feasibility (including trained personnel, laboratory capacity for resistance testing and virus screening, and frequent maintenance of traps). For egg collection, a large number of sites were selected since the monitoring was delegated to trained local agents. However, a limited number of localities were selected for adult collection because the activities were conducted by entomological teams from the Institut Pasteur de Dakar (IPD). The selected localities also reflect diverse urban typologies and ensure geographic and ecological representativeness across the three main administrative districts of Dakar (Dakar district: Médina, Ouakam, Point E, and Grand Yoff - Guédiawaye district: Guédiawaye - Pikine district: Mbao). Grand Yoff and Guédiawaye are lively, densely populated areas with informal housing. Mbao blends urban life with semi-rural and industrial zones. Médina, Dakar's oldest neighborhood, is culturally rich but crowded. Ouakam combines a traditional fishing village with modern developments. Point E is an upscale area with reliable infrastructure.

Host-seeking mosquitoes were collected using Biogents (BG-Sentinel) traps baited with a chemical lure (BG-Lure: attractant designed to mimic human skin odors) and $CO_2$. The lure was replaced every four weeks in accordance with the manufacturer's recommendations to maintain effectiveness during the study period.

In each locality, one house was selected among those where ovitraps were set up. Two traps were deployed (one indoor and one outdoor) in each house for three consecutive days. Collection bags were retrieved and replaced twice daily early in the morning at 07:00 a.m. and 07:00 p.m., to separate day and night collections. The outdoor traps were positioned in shaded and sheltered areas such as covered verandas or courtyards, protected from direct sun and rain, to optimize trap efficiency and mimic natural environments for *Ae. aegypti*. However, the indoor traps were placed in the bedrooms. Resting mosquitoes were sampled using a Prokopack aspirator in three houses per locality. Collections were conducted at equal time for 10 min indoor (bedrooms) and outdoor (semi-enclosed spaces such as corridors and courtyards within the house compound).

After collection, adult mosquitoes were immobilized on ice in the field. Mosquitoes were then transported in cool boxes to the laboratory, where they were killed and stored at −80 °C until further processing. Morphological identification was performed on a chill table to prevent RNA degradation. All mosquitoes collected were identified using standard morphological keys [21,22] and then pooled according to species, sex, feeding status (unfed and blood-fed), locality, method and date of collection. Number of unfed *Ae. aegypti* per pool ranging from 1 to 20 maximum while *Ae. aegypti* blood-fed was pooled individually. and 160 for the other species. Only *Ae. aegypti*, the unique vector identified was tested for virus detection. Other mosquito species were identified and pooled to document the biodiversity and for potential future analyses. Each pool of *Ae. aegypti* mosquitoes (whole mosquitoes for non-engorged and heads and thoraxes for engorged females) was homogenized in L-15 medium (GibcoBRL, Grand Island, NY, USA) for RNA extraction and arbovirus detection. For engorged females, the abdomens were homogenized in Phosphate-Buffered Saline (PBS), and the homogenate was used to determine the origin of their blood meals.

## Detection of arboviruses in *Aedes aegypti* by real-time RT-PCR

To screen for arboviruses (*Flaviviruses* and *Alphaviruses*), *Ae. aegypti* pools were homogenized in 1.5 ml cryogenic vials containing 500 µl of L-15 medium (GibcoBRL, Grand Island, NY, USA) using sterile pestles in a biosafety level 2 laboratory. Homogenates were centrifuged at 8,000 rpm for 10 minutes at 4°C to remove mosquito debris. These were subsequently combined into super pools (collecting 50 µl of supernatant per pool) stratified by physiological status (blood-fed vs. unfed) and sex. Each super pool combined 12–14 pools contained 600–700 µl of supernatant for RNA extraction. After RNA extraction, 100 µl of supernatant from each super pool was processed using the QiaAmp Viral RNA Extraction Kit (Qiagen, Heiden, Germany), following the manufacturer's instructions.

After viral RNA extraction, the super pools were tested for pan-*Flaviviruses* and pan-*Alphaviruses* using real-time quantitative reverse transcription polymerase chain reaction (qRT-PCR) on a Bio-Rad CFX96 instrument with SYBR Green-based melting curve analysis. Detection of *Alphaviruses* was performed as described in Vina-Rodriguez A et al. (2017), with a slight modification [23]. The QuantiTect SYBR Green RT-qPCR kit (QIAGEN, Hilden, Germany) was used for amplification, with dengue and chikungunya viruses serving as positive controls (Cq ≈ 28), supplied at the Centre de Recherche sur les Arbovirus (CRORA) biobank [24]. Pools within each positive super pool were subsequently tested individually for *Flavivirus* and *Alphavirus.* Furthermore, sequencing was performed on positive pools to identify specific viruses. Male mosquitoes were included in the screening to detect potential vertical transmission of arboviruses. For females, fed and unfed were pooled separately to detect infected (maybe infectious) vs infectious females.

## Next-generation sequencing and genome assembly

RNA extracts from *Flaviviruses*-positive mosquito pools were processed to obtain complete genomes using a target enrichment standard hybridization approach with the Twist Biosciences Comprehensive Viral Research Panel (CVRP) [25]. This sequencing was conducted for phylogenetic analysis and determine the genetic diversity. The enriched sample libraries were prepared according to Twist Technical Support recommendations and subsequently loaded onto an Illumina iSeq 100 sequencing system, following the manufacturer's guidelines. Raw sequencing data were generated in fastq format, and genome assembly was performed using the CZ-ID platform, an open-source metagenomics tool available at https://www.twistbioscience.com/sites/default/files/resources/2020-11/ProductSheet_NGS_ComprehensiveViralResearchPanel_11Nov20_Rev1.0.pdf, as already described [26,27].

## Phylogenetic analysis

Sequence alignment was performed using MAFFT [28], and the alignment was further processed with IQ-TREE [29]. A maximum-likelihood (ML) phylogenetic tree was then constructed and visualized with Figtree V1.4.4 [30].

## Blood feeding patterns of *Aedes aegypti*

The blood-fed female mosquitoes used to calculate the anthropophilic rate (AR) were collected using both BG-Sentinel traps and aspiration, as described above.

The origin of blood meals from engorged female mosquitoes of *Ae. aegypti* was determined using a modified Enzyme-Linked ImmunoSorbent Assay (ELISA) protocol originally described by Beier et al. [31]. Polyclonal anti-IgG antibodies were used to detect host-specific immunoglobulins from vertebrate species commonly present in urban environments, including humans, pigs, horses, sheep, cats, dogs, rats, cattle, rabbits, and chickens.

## Insecticide resistance assessment in *Aedes aegypti* samples from Dakar

Insecticide susceptibility tests were conducted following WHO guidelines [32] on 3–5-day-old non-blood-fed F1 *Ae. aegypti* females. The eggs (F0) used for these tests come from the ovitraps described above. In order to limit the

presence of many inbreeding in the populations tested, eggs were collected over several weeks and were mixed. This method ensures that the females tested represent a genetically diverse sample. Eggs were reared under laboratory (controlled conditions) until adult emergence. These adults were then allowed to mate, and the resulting F1 generation was used for the bioassays. The bioassays were performed under controlled conditions: 25°C (± 2°C) and 70% (±10%) relative humidity. Four replicates of 20–25 females per tube were exposed to each insecticide for 1 hour. The Vector Control Research Unit, Universiti Sains Malaysia (WHO collaborating Centre) provided all insecticide-impregnated papers used in this study. Tested insecticides papers included: pyrethroids (0.4% and 3.75% permethrin, 0.03% and 0.25% deltamethrin, 0.08% and 0.25% lambda-cyhalothrin, 0.08% and 0.25% alpha-cypermethrin), carbamates (0.2% bendiocarb) and organochlorines (1.5% malathion, 60 mg/m2 pirimiphos-methyl and 1.0% chlorpyriphos-ethyl). The lower concentrations used correspond to the WHO-recommended new diagnostic doses for assessing *Aedes* susceptibility. The higher concentrations (pyrethroids 3.75% permethrin, 0.25% deltamethrin 0.25% lambda-cyhalothrin and 0.25% alpha-cypermethrin) are not currently the standard for *Aedes* but are usually applied in *Anopheles* intensity essay. They were used here due to the availability of pre-impregnated papers and to provide initial data one the populations under study. Papers impregnated with olive oil (for organophosphate and carbamate control), and silicone oil (for pyrethroid control) were used as control. For pyrethroids, cumulative knockdown rates were recorded after 10, 20, 30, 40, 50, 60, 70 and 80 min. After 1-hour exposure period, mosquitoes were transferred to holding tubes, provided with 10% sugar solution, and kept at 25°C (± 2°C) and 70% (±10%) relative humidity. Mortality was recorded at 24-hours post-exposure.

## Genotyping of *kdr* mutations in *Aedes aegypti* populations

A subsample of 1,014 *Aedes* mosquitoes, phenotypically classified as resistant or susceptible to insecticides (0.4% permethrin, 0.03% deltamethrin, 0.08% lambda-cyhalothrin, 0.08% and alpha-cypermethrin) based on bioassay results, was randomly selected for genotyping of the *kdr* mutations F1534C, S989P, V1016I and V1016G. A maximum of 26 mosquitoes per insecticides (alive or dead) were genotyped. When fewer than 26 mosquitoes were available, all were included in the analysis. A total of 536 resistant *Ae. a*egypti mosquitoes and 479 susceptible mosquitoes representing mosquitoes from all the six study neighborhoods were used for the genotyping. Total DNA was extracted from whole mosquitoes using the CTAB 2% [33]. *Kdr* mutations were detected using allele-specific PCR according to the protocols of Saavedra-Rodriguez et al [34] and Bariami et al [35]. The single 10 μL volume reaction contained 2.5 μL of DNA sample, 5 μL of Luna Universal qPCR Master Mix (SuperMix), 1.3 μl of sterile water, 0.4 of Probe and 0.4 μL of each primer. The cycling conditions were: 55° C for 10 min, followed by 95° C and 40 cycles of 95° C for 60 s, 60° C for 30 s.

## Statistical analysis

The relative abundance of different mosquito species was calculated per locality. The distribution and abundance of *Ae. aegypti* in the study area were assessed using the ovitrap positive index (OPI; number of ovitraps containing *Aedes* eggs/ number of traps examined × 100) and egg density index (EDI; average number of *Aedes* eggs per ovitrap). The anthropophilic rate (AR; number of mosquitoes that fed on human blood/ number of all fed mosquitoes x100), outdoor biting rate (OBR; number of mosquitoes collected host-seeking outdoor/ total number of mosquitoes collected host-seeking x 100) and endophilic rate (ER; number of mosquitoes collected by resting indoor/ total number of mosquitoes collected by aspiration x 100) were calculated for *Ae. aegypti* for each site. The effect of sampling sites on the species richness and EDI of *Ae. aegypti* was analyzed using a generalized linear mixed-effect model, using sampling periods as random factors, with Poisson error distribution. Tukey's honestly significant difference (HSD) post hoc test was used to identify significant pairwise comparisons.

Categorical variables (OPI, OBR, AR, and ER) of all sites were first compared using Chi-square or Fisher exact test when appropriate. When the test was statistically significant, sites with the lowest values were progressively removed until the remaining become comparable.

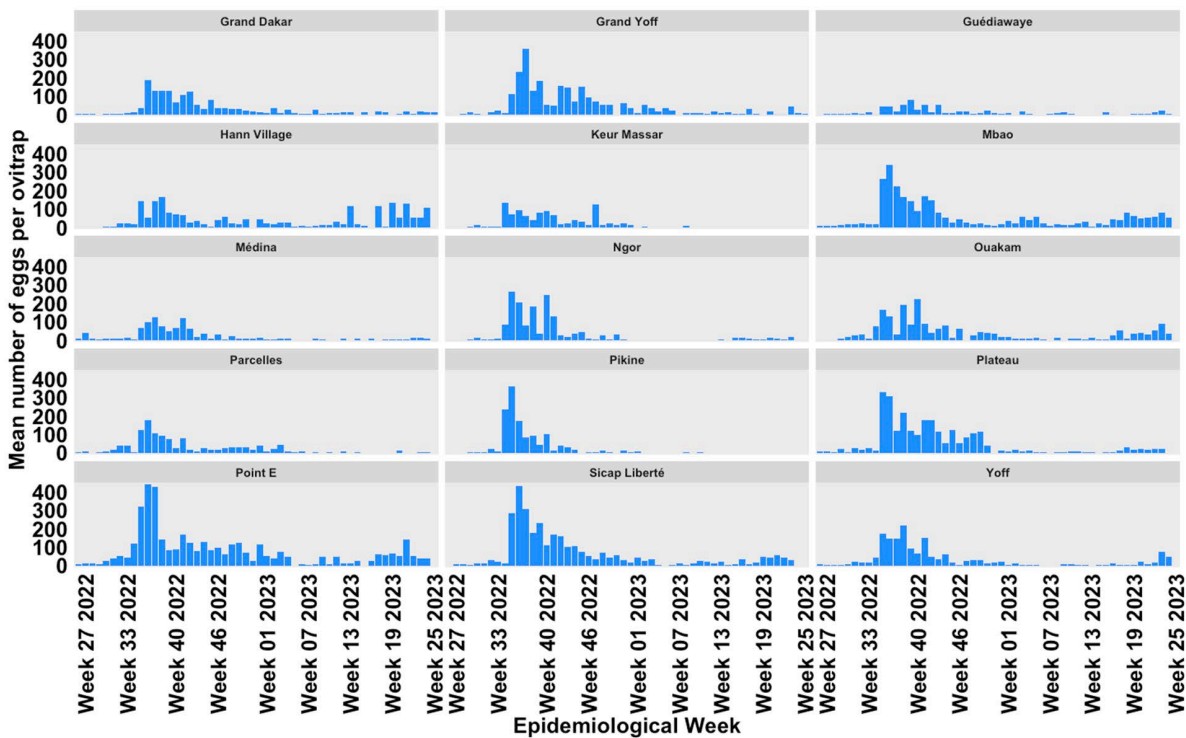

PLOS Neglected Tropical Diseases

Mosquito susceptibility to insecticides was interpreted according to WHO guidelines [32]. Mortality rates between 98–100% indicated susceptibility, mortality rates from 90 to 97% indicated possible resistance and mortality rates less than 90% indicated confirmed resistance. The Abbott formula was applied for mortality correction when necessary [32].

Knockdown times ($KDT_{50}$ and $KDT_{95}$) were calculated using the Probit model with a 95% confidence interval using the *BioRssay* package [36]. *Kdr* genotype frequencies were compared between phenotypically resistant (alive) and susceptible (dead) mosquitoes using Pearson's chi-square test or Fisher's exact test, depending on sample size. All tests were considered statistically significant when $p < 0.05$. Data were analyzed using R statistical software, version 4.3.1 [37].

## Results

### Spatio-temporal distribution of *Aedes* species eggs during the study period

A total of 282,534 *Aedes* eggs were collected during the study period, with an average of 39.12 eggs per ovitrap per week. The number of eggs per ovitrap ranged from 0 to 1,497. *Aedes* eggs were recorded in all sites throughout most of the sampling period.

The highest EDI were observed between August to October (epidemiological weeks 34–43) in all study sites (Fig 2). Thereafter, egg density gradually declined over the following weeks.

A significant variation in the OPI was observed across the different neighborhoods (Table 1; $\chi^2 = 486.44$, $df = 14$, p-value < 2.2e-16). The highest and statistically comparable OPI were recorded in Sicap Liberté (59.8), Point E (58.04) and Mbao (57.84), whereas the lowest values were observed in Pikine, Guédiawaye and Keur Massar.

The EDI of *Ae. aegypti* was similar only between Keur Massar and Medina ($p = 0.09$), and between Ngor and Yoff ($p = 0.99$). All other pairwise comparisons showed statistically significant differences ($p < 0.001$).

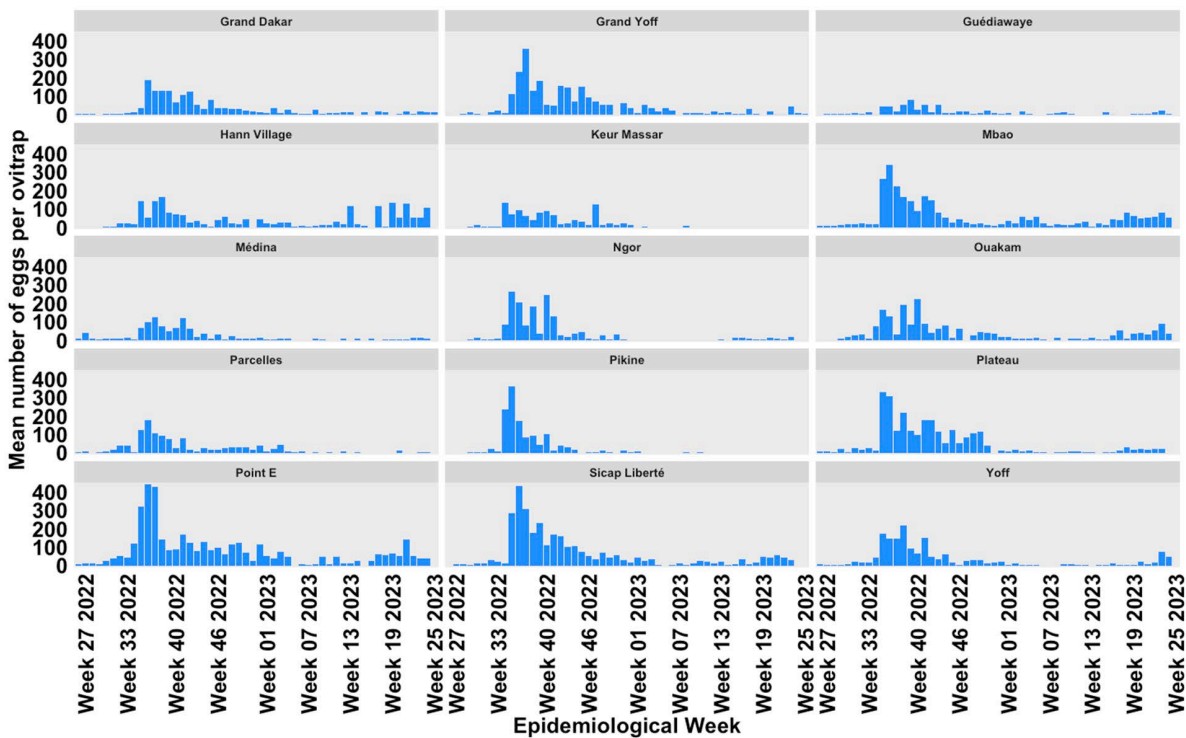

**Fig 2. Spatial distribution and frequency of *Aedes* eggs collected in Dakar during the surveillance period, july-2022 to August 2023.**

**Table 1. Ovitrap indices in 15 localities of Dakar from July 2022 to August 2023.**

| Sites | Ovitraps placed | Positive ovitraps | Ovitrap positive index (OPI) | Egg Numbers | Egg density index (EDI) |
|---|---|---|---|---|---|
| Grand Dakar | 510 | 175 | 34.31 | 15492 | 16.86 [10.84-26.2] |
| Grand Yoff | 500 | 209 | 41.80 | 23974 | 27.20 [17.49-42.3] |
| Guédiawaye | 510 | 130 | 25.49 | 6237 | 6.78 [4.36-10.6] |
| Hann Village | 460 | 203 | 44.13 | 20212 | 22.52 [14.48-35.0] |
| Keur Massar | 469 | 109 | 23.24 | 9777 | 11.01 [7.08 -17.1] ** |
| Mbao | 510 | 295 | 57.84* | 28251 | 30.72 [19.76 -47.8] |
| Médina | 520 | 174 | 33.46 | 9715 | 10.52 [6.76-16.4] ** |
| Ngor | 420 | 136 | 32.38 | 14891 | 18.00 [11.58-28.0] * |
| Ouakam | 489 | 208 | 42.54 | 19324 | 21.53 [13.85-33.5] |
| Parcelles | 510 | 150 | 29.41 | 11539 | 12.55 [8.07-19.5] |
| Pikine | 350 | 93 | 26.57 | 12569 | 15.85 [2.47-3.06] |
| Plateau | 490 | 200 | 40.82 | 25505 | 28.68 [18.45-44.6] |
| Point E | 510 | 296 | 58.04* | 38720 | 42.31 [27.22-65.8] |
| Sicap Liberté | 490 | 293 | 59.80* | 30252 | 34.09 [21.93- 53.0] |
| Yoff | 508 | 191 | 37.60 | 16076 | 17.76 [11.42-27.6]* |

(*) Stars indicate that the EDI and the OPI values for these sites are not significantly different.

## Adult mosquito species abundance

A total of 34,666 mosquitoes belonging to four genera (*Aedes, Culex, Mansonia and Anopheles*) and 16 species were collected. Species richness varied across the surveyed localities, ranging from 3 species in Ouakam to 15 species in Mbao (S1 Table). Mbao exhibited the significantly highest mean species richness (Mean = 6.31; 95% CI: 4.26–9.34) compared to the other localities (p < 0.05). The other localities showed lower and comparable mean species richness (S2 Table). The most abundant species were: *Cx. quinquefasciatus* (n = 30,121; 86.9% of the mosquito fauna), *Ae. aegypti* (n = 1959; 5.65%), *Ma. africana* (n = 1037; 3.0%), *Cx. tritaeniorhynchus* (n = 712; 2.0%), *Ma. uniformus* (n = 435; 1.25%) and *An. gambiae* s.l. (n = 298; 0.8%). Each of the other species (*An. pharoensis, An. sp, An. rufipes, Cx. theileri, Cx. cinereus, Cx. antennatus, Cx. neavei, Cx. nebulosus, Cx. perfuscus, Cx. poicilipes*) represented less than 0.5% of the total collection.

## *Aedes aegypti* adult mosquito abundance

*Aedes aegypti* was the only species of the *Aedes* genus collected, and its presence was recorded at all study sites (Table 2). The relative abundances of this species across sites showed a statistically significant difference (χ2 = 10.86, df = 3, p = 0.01). The proportions observed in Grand Yoff (7.50%), Guédiawaye (1.04%), Mbao (1.21%) and Ouakam (4.45%) were statistically comparable (χ2 = 5.19, df = 3, p = 0.16) and consistently lower than those in Point E (36.11%) and Médina (19.83%).

## Resting and biting patterns

*Ae. aegypti* females were primarily collected outdoors in three localities (Grand Yoff, Mbao and Ouakam), while in the other localities, they were collected both indoors and outdoors. The OBR varied significantly across the six sites (p = 0.0004). The OBR of Mbao, Grand Yoff and Ouakam were higher and comparable (p = 0.08). Host-seeking activity was mainly diurnal, with 68.5% of the females collected indoors during the daytime and 56.5% outdoors during the daytime. A higher proportion of female *Ae. aegypti* was found resting indoors in only one of the six investigated neighborhoods (Fig 4). The ER of *Ae.*

*aegypti* varied significantly across the six sites (p = 0.04). The ER at Point E was higher than the five others, which were comparable (p = 0.16).

## Host preferences of adult *Aedes aegypti*

The tested females exhibited a strong Anthropophilic rate, which accounted for 75% of the identified blood meals (105 of 140) (Fig 5). The AR varied significantly across sites (p = 0.0008). The AR at Médina was lower than the other sites wish were comparable (p = 0.7). Other hosts detected included cattle, horses and sheep.

## Virus detection and serotyping

Virus detection assays were conducted on 1959 (616 pools) *Ae. aegypti* mosquitoes, including 874 females (395 pools) and 1,085 males (221 pools) (Table 3) and combined into super pools.

**Table 2. Number of adult *Aedes aegypti* collected from August 2022 to July 2023 in Dakar.**

| | Aspiration | | | BG Lure + CO2 | | | | | | Total |
|---|---|---|---|---|---|---|---|---|---|---|
| | Indoor | Outdoor | | Indoor | | Outdoor | | | | |
| Localities | | | Total | Day | Night | Day | Night | Total | | |
| Grand Yoff | 38 | 73 | 111 | 5 | | 15 | 6 | 26 | | 137 |
| Guédiawaye | 21 | 45 | 66 | 16 | 12 | 26 | 10 | 64 | | 130 |
| Mbao | 27 | 86 | 113 | 6 | 2 | 18 | 23 | 49 | | 162 |
| Médina | 34 | 151 | 185 | 45 | 15 | 47 | 29 | 136 | | 321 |
| Ouakam | 12 | 10 | 22 | 4 | 5 | 50 | 30 | 89 | | 111 |
| Point E | 151 | 201 | 352 | 270 | 125 | 186 | 165 | 746 | | 1098 |
| **Total** | **283** | **566** | **849** | **346** | **159** | **342** | **263** | **1110** | | **1959** |

In all neighborhoods the highest densities of *Ae. aegypti* adults were recorded from August to November, corresponding to the rainy season (Fig 3). Overall vector abundance peaked in September, then declined gradually, reaching very low levels between December and June.

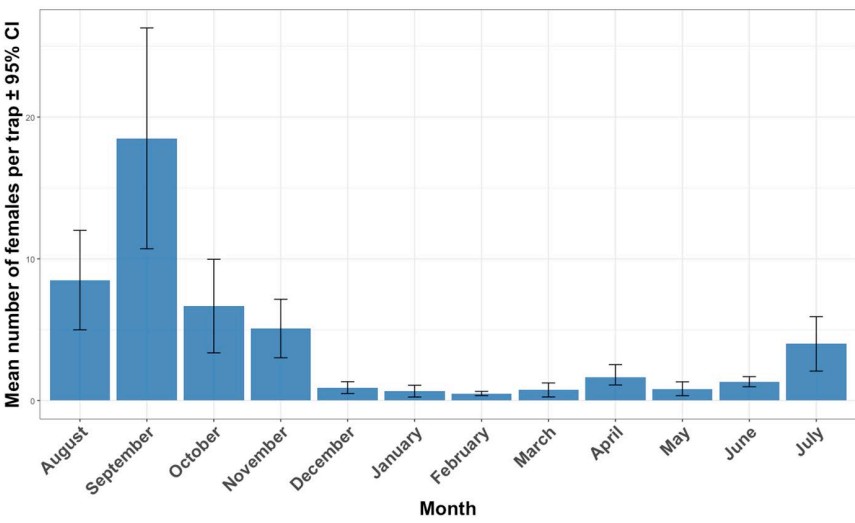

**Fig 3. Mean number of host-seeking *Aedes aegypti* females collected using Biogents (BG) Sentinel traps in Dakar from August 2022 to July 2023.** Error bars indicate 95% CI: 95% Confidence interval.

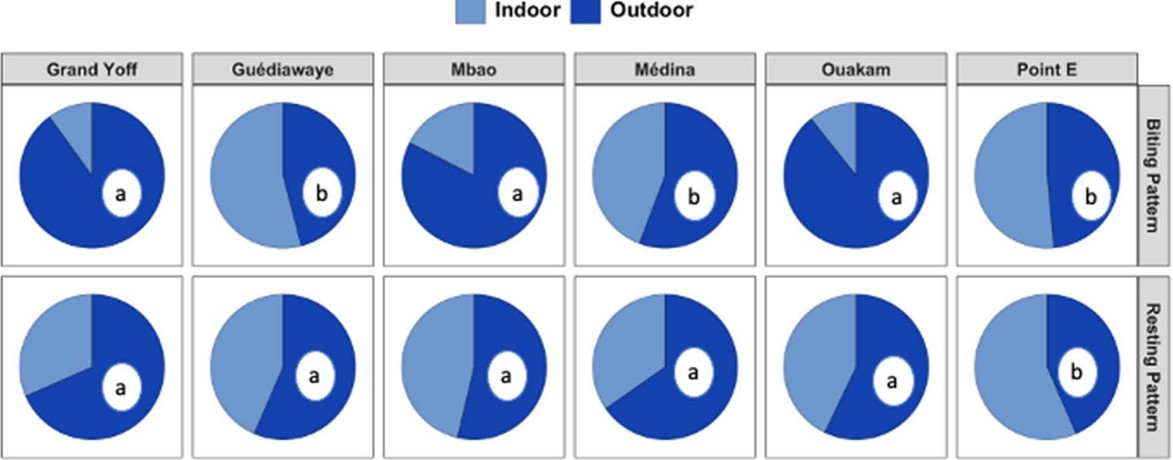

**Fig 4. Indoor and outdoor host seeking and resting behavior of adult female *Aedes aegypti* collected in in six localitiesin Dakar from August 2022 to July2023.** For each variable, localities with different letters are significantly differents.

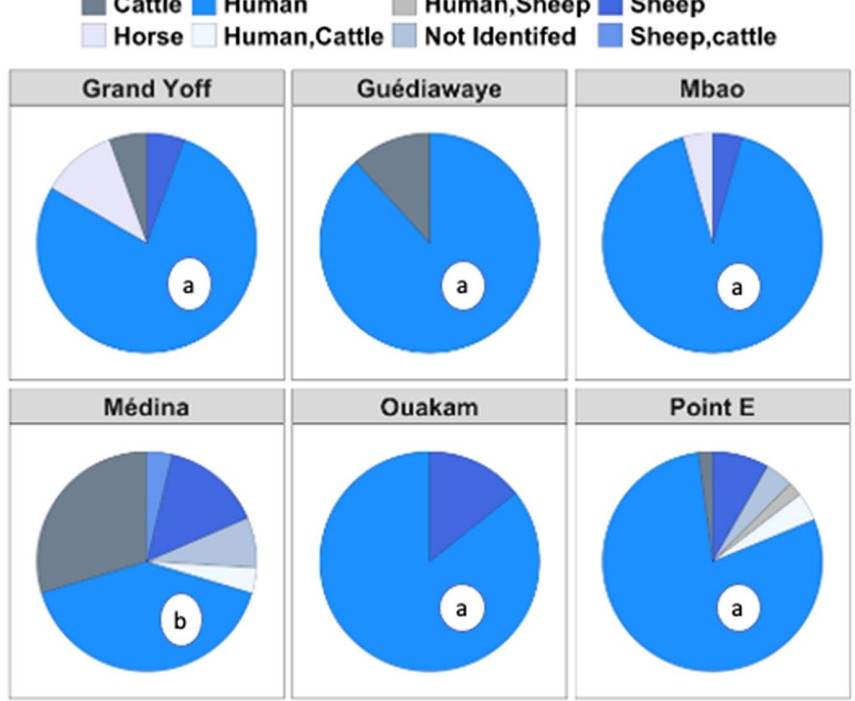

**Fig 5. Blood feeding patterns of *Aedes aegypti* collected in six neighborhoods of Dakar, August 2022 to July 2023.** Anthropophilic rates with different letters are significantly different.

Overall, four pools were positive for *Flaviviruses* by qRT-PCR (Table 3). No pool was positive for *Alphaviruses*. All positive pools were blood-fed females that were collected in September (Guédiawaye, 1 pool), November (Grand Yoff and Mbao, 1 pool each) and December 2022 (Point E, 1 pool). Of these, only two samples were successfully serotyped as

**Table 3. *Aedes aegypti* mosquitoes positive for arboviruses in Dakar, August-2022 to July 2023.**

| Sites | Number of Mosquitos | Pools positive *Flaviviruses* | Month of collection | Serotype |
|---|---|---|---|---|
| Grand Yoff | 137 | 1 | November | |
| Guédiawaye | 130 | 1 | September | Dengue-3 |
| Mbao | 162 | 1 | November | |
| Médina | 321 | 0 | | |
| Point E | 1098 | 1 | December | Dengue-3 |
| Ouakam | 111 | 0 | | |
| Total | 1959 | 4 | | |

DENV-3. The sequences obtained in this study were analyzed alongside a subset of DENV whole genomes from various serotypes and genotypes available in GenBank. Phylogenetic analysis revealed that the detected DENV-3 strain belongs to the genotype III clade (MP621_Senegal = PV834968; MP638_Senegal = PV834969). These sequences clustered in a distinct clade, forming a monophyletic group with strains previously reported in Senegal (2018), Burkina Faso (2017) and Benin (2023) (Fig 6).

### Insecticide susceptibility status of *Aedes aegypti* populations

Results of insecticide susceptibility tests on *Ae. aegypti* populations collected from six neighborhoods in the Dakar region are presented in Fig 7. In all tests, no mortality was observed in the control groups.

All tested *Ae. aegypti* populations showed high resistance to diagnostic doses of pyrethroids, with mortality rates ranging from 35.35% to 64.63%. Most of these populations also exhibited resistance to *Anopheles* intensity doses. Suspected resistance was noted with 0.25% deltamethrin in 3 localities (Ouakam: 93%, Point E:92.55% and Mbao: 92.10%), 0.25% lambda-cyhalothrin in 2 localities (Guédiawaye: 94.68% and Mbao: 92.38%), and 0.25% alpha-cypermethrin in 2 localities (Mbao: 97% and Guédiawaye: 90.62%). Only the population of Ouakam showed sensitivity to 0.25% lambda-cyhalothrin (97.84%), while *Ae. aegypti* populations from Point E (94.68%) showed moderate resistance to 3.75% permethrin (94.68%). In other localities, mosquitoes were susceptible.

The same observation was made for the organophosphates, where all mosquito populations exhibited complete resistance, except for those in Guédiawaye (90%) and Point E (96%), which showed suspected resistance to malathion and pirimiphos-methyl, respectively.

The Knockdown times ($KDT_{50}$ and $KDT_{95}$) values at diagnostic doses were considerably higher compared to those observed at higher-intensity pyrethroids doses in all *Ae. aegypti* populations tested (S3 Table), indicating a strong level of phenotypic resistance. At standard diagnostic doses, mosquitoes required more time to be knocked down, values frequently exceeding one hour (Fig 8).

### Genotyping of *kdr* resistant mutations and their association with phenotypic resistance

Genomic DNA of 1,014 individual mosquitoes, including alive and dead specimens exposed to pyrethroids, was obtained from six neighborhoods. Assays were then performed to detect the S989P, V1016G/I and F1534C mutations. The genotypes and allele frequencies of each *kdr* mutation are shown in Table 4 and S4 Table.

Only two homozygous mutation types were observed in the tested samples: 1534C and 989P. The 1534C mutation was detected at all sites except Médina and Guédiawaye in both resistant (allelic frequencies: 0 to 0.50) and susceptible groups (allelic frequencies: 0 to 0.54). The allele frequency for the 989P mutation ranged from 0 to 0.52 in the resistant group and 0 to 0.53 in the susceptible group in the localities of Médina, Point E and Guédiawaye. The predominant genotype was the heterozygous mutant for the F1534C, S989P and V1016G mutations. The V1016I mutation was not

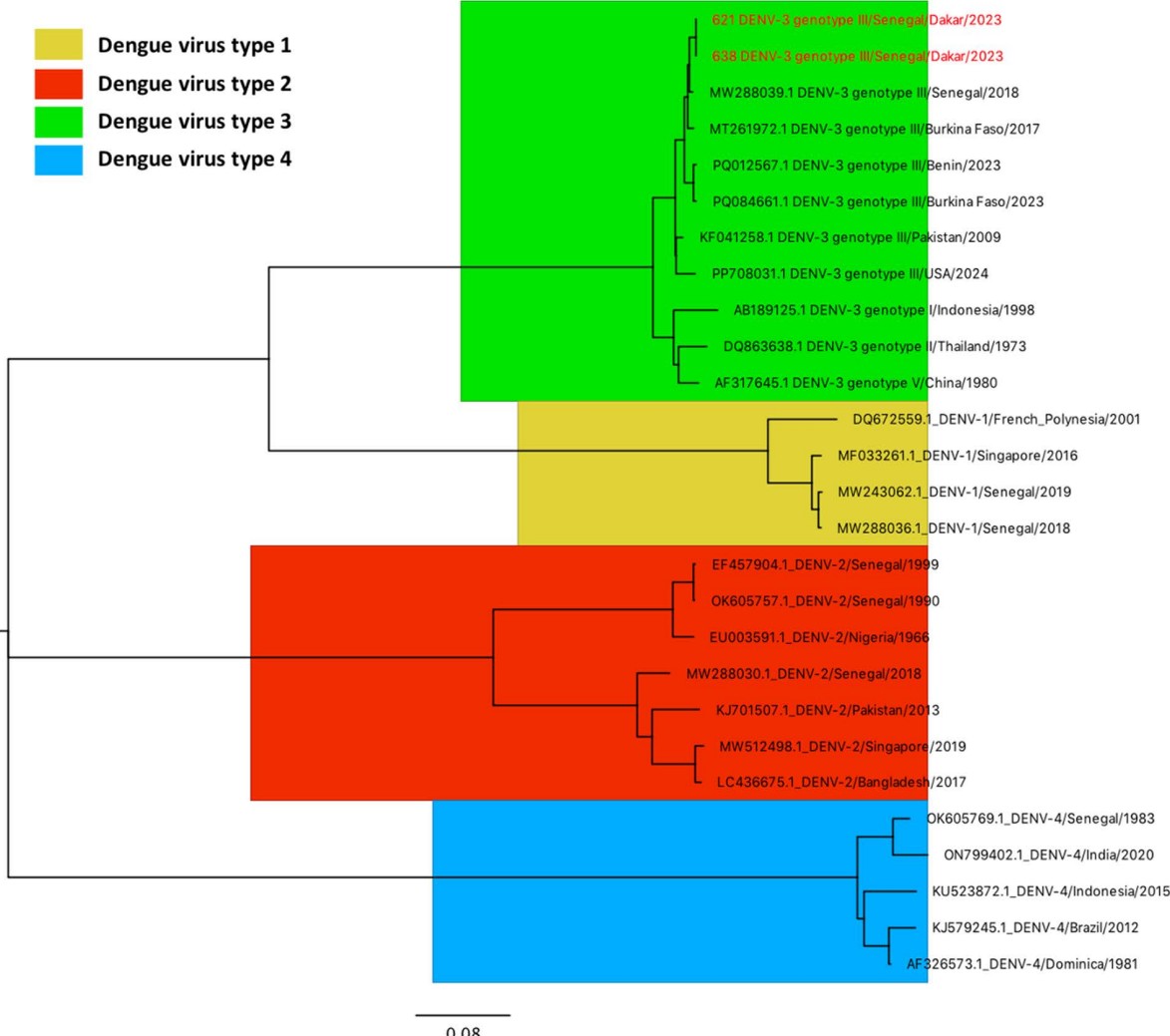

**Dengue virus type 1**
**Dengue virus type 2**
**Dengue virus type 3**
**Dengue virus type 4**

621 DENV-3 genotype III/Senegal/Dakar/2023
638 DENV-3 genotype III/Senegal/Dakar/2023
MW288039.1 DENV-3 genotype III/Senegal/2018
MT261972.1 DENV-3 genotype III/Burkina Faso/2017
PQ012567.1 DENV-3 genotype III/Benin/2023
PQ084661.1 DENV-3 genotype III/Burkina Faso/2023
KF041258.1 DENV-3 genotype III/Pakistan/2009
PP708031.1 DENV-3 genotype III/USA/2024
AB189125.1 DENV-3 genotype I/Indonesia/1998
DQ863638.1 DENV-3 genotype II/Thailand/1973
AF317645.1 DENV-3 genotype V/China/1980
DQ672559.1_DENV-1/French_Polynesia/2001
MF033261.1_DENV-1/Singapore/2016
MW243062.1_DENV-1/Senegal/2019
MW288036.1_DENV-1/Senegal/2018
EF457904.1_DENV-2/Senegal/1999
OK605757.1_DENV-2/Senegal/1990
EU003591.1_DENV-2/Nigeria/1966
MW288030.1_DENV-2/Senegal/2018
KJ701507.1_DENV-2/Pakistan/2013
MW512498.1_DENV-2/Singapore/2019
LC436675.1_DENV-2/Bangladesh/2017
OK605769.1_DENV-4/Senegal/1983
ON799402.1_DENV-4/India/2020
KU523872.1_DENV-4/Indonesia/2015
KJ579245.1_DENV-4/Brazil/2012
AF326573.1_DENV-4/Dominica/1981

0.08

**Fig 6. Phylogenetic relationships between newly generated DENV whole-genome sequences from Dakar (collected between August 2022 to July 2023) and reference sequences representing various serotypes and genotypes retrieved from GenBank, including previously reported West African genotypes.**

detected. In the main analysis, no significant association was observed between *kdr* mutations and mosquito phenotype (alive or dead), except in Guédiawaye, where the frequencies of S989P and V1016G were significantly higher in the resistant group (Table 4).

Thirteen genotypes were observed across the four *kdr* loci in the 1,014 mosquitoes genotyped (Fig 9). The most common four-locus genotype detected across all sites were the heterozygous mutants for F1534C (FC), S989P (SP) and V1016G (VG) combined with the homozygous susceptible type for V1016I (VV). This four-locus genotype (FF/VV/VG/SP) was found in 295 (29.20%) resistant and 277 (27.42%) susceptible specimens. The four-homozygous mutant FC/VV/VG/SP was present in 159 (15.74%) resistant and 135 (13.36%) susceptible specimens (Fig 9). Multi-locus genotypes combining these mutations were detected at comparable frequencies in both resistant and susceptible mosquitoes in all localities (p > 0,05).

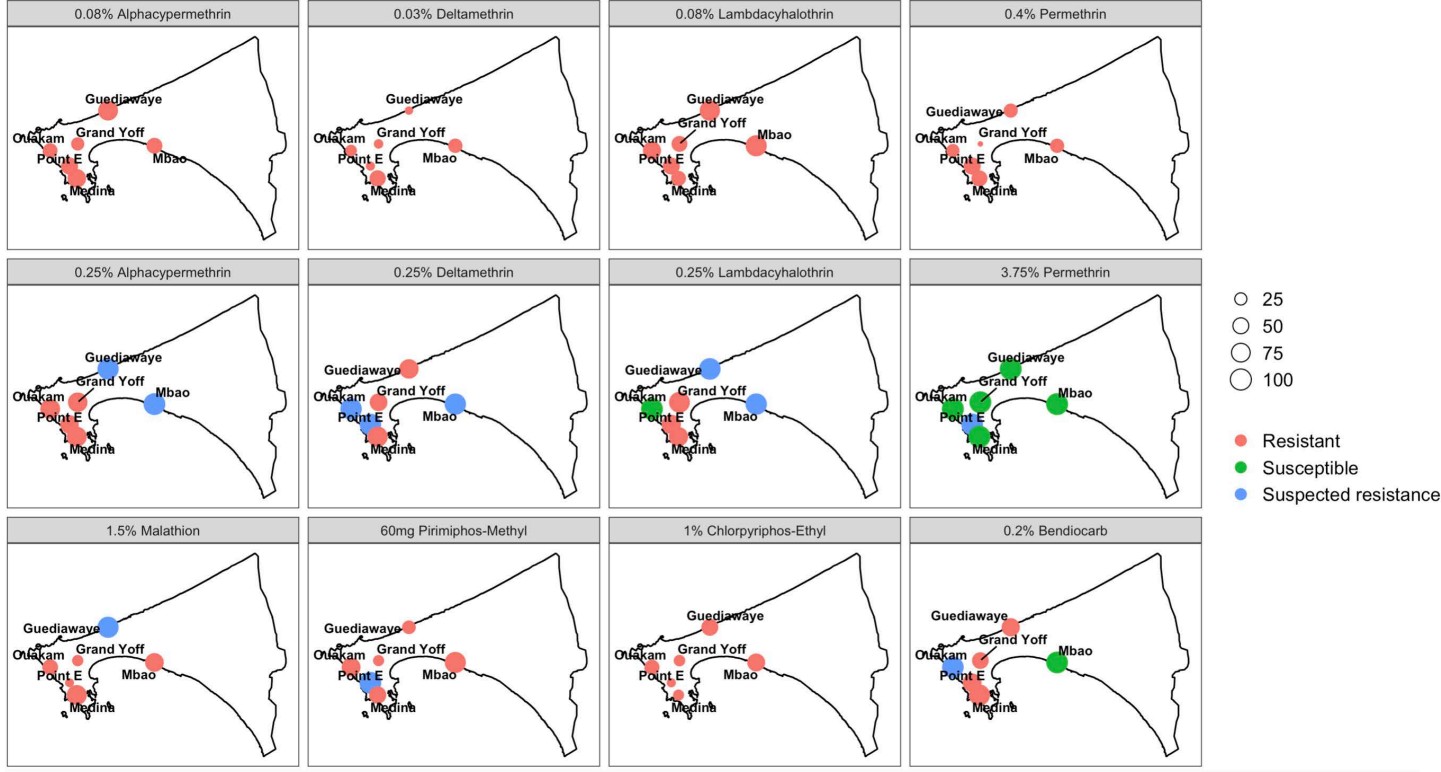

**Fig 7. Insecticide susceptibility status of *Aedes aegypti* populations collected from six localities in Dakar from August 2022 to July 2023.** This map was created using the R software (version 4.4.0) and the package raster using an empty shapefile from the HDX website (https://data.humdata.org/dataset/senegal-administrative-boundaries) available under Creative Commons Attribution 4.0 International licence.

## Discussion

Long-term vector surveillance is crucial for developing effective strategies to prevent and control dengue transmission. This study provides valuable information on the spatiotemporal distribution and population density of *Aedes* vectors, based on egg and adult collections, as well as evidence of insecticide resistance and associated *kdr* mutations. It represents the first longitudinal assessment of *Ae. aegypti* populations in Dakar, the capital of Senegal.

Our findings indicate that, *Ae. aegypti* eggs were present in high numbers across all study areas during most of the sampling periods. The use of ovitraps allowed for the identification of varying infestation levels and seasonal fluctuations in vector populations [38]. The detection of eggs confirms the presence of gravid females having actively fed on vertebrate hosts, underscoring the risk of arbovirus transmission. Thus, ovitraps play a key role in enhancing vector surveillance and enabling early detection of vector presence [39–41]. In this study, ovitraps were used as a sensitive monitoring tool to detect *Aedes* presence and estimate population density. It was observed that, the proportion of *Aedes* eggs was significantly higher in Mbao, Point E, and Sicap Liberté compared to other areas. Notably, the highest egg density was recorded in September (rainy season), coinciding with the peak period of DENV-positive mosquito detections (August-December: rainy season). These localities and periods should be prioritized for vector control operations. The use of ovitraps have been shown in other contexts to potentially reduce vector populations highlighting their sensitivity as a vector control tools [42,43]. Although beyond the scope of this study, the evaluation of their impact on mosquito control should be explored in our context.

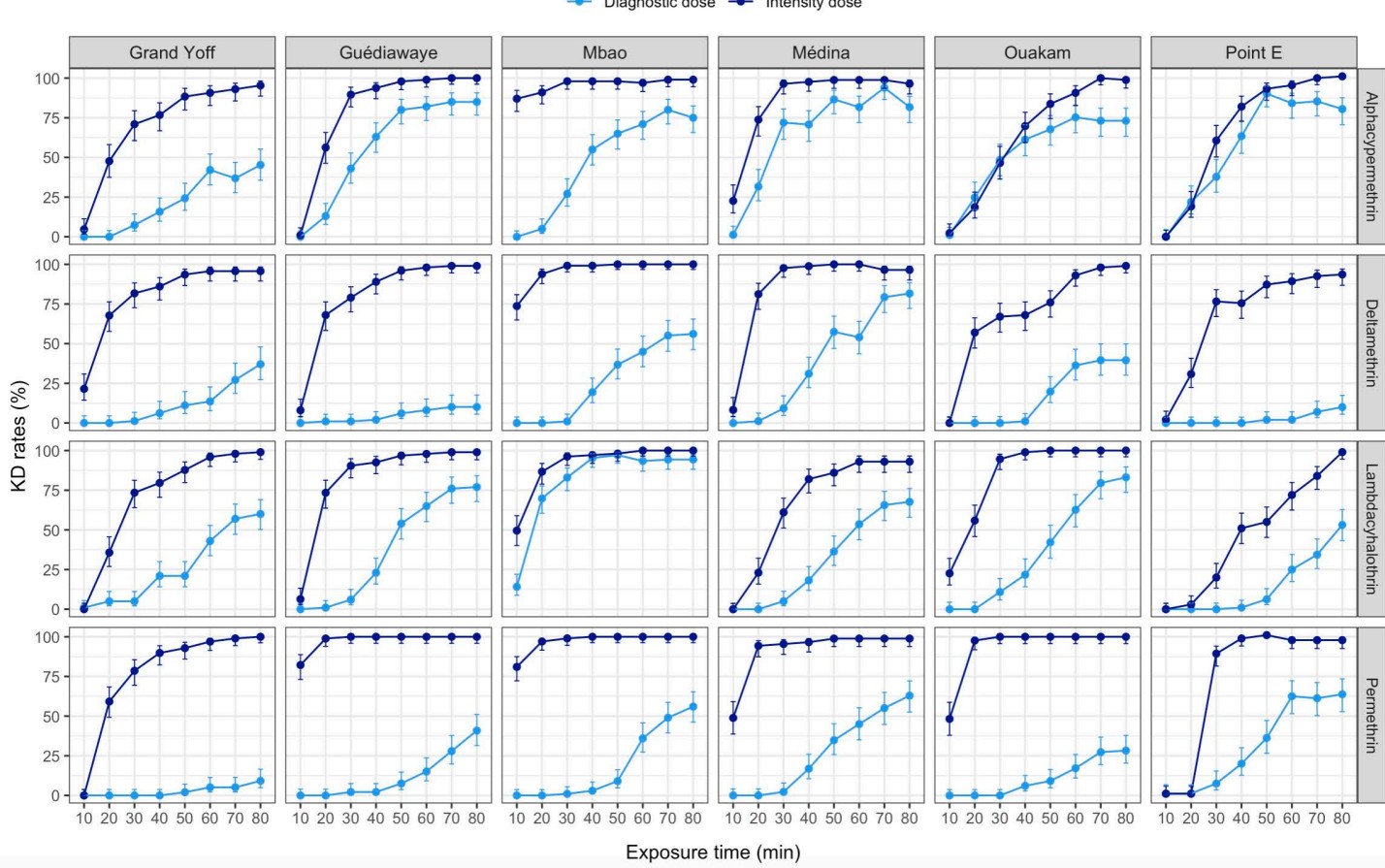

**Fig 8. Knockdown rates of *Aedes aegypti* from six localities in Dakar following exposure to pyrethroids from August 2022 to July 2023.**
KD = Knockdown, Diagnostic dose = 0.4% permethrin, 0.03% deltamethrin, 0.08% lambda-cyhalothrin, 0.08% alpha-cypermethrin, Intensity dose = 3.75% permethrin, 0.25% deltamethrin, 0.25% lambda-cyhalothrin, 0.25% alpha-cypermethrin.

This study also revealed variations in species richness and abundance across the sites. Higher species diversity was observed in Mbao, a site near a classified forest, whereas lower species diversity and higher mosquito abundance were found in the two most urbanized localities (Point E and Médina). This is in accordance with a previous study that suggested that human-induced environmental changes tend to favor mosquito abundance while reducing species diversity, whereas natural environments typically support higher species diversity but lower abundance [44].

Blood meal analysis indicated that, *Ae. aegypti* predominantly fed on humans in most areas except Médina, indicating a high risk of arbovirus transmission [45]. We observed highly anthropophilic behavior in *Ae. aegypti*, the main vector of DENV was consistent with the results of previous studies conducted in Senegal [1,46]. However, the host diversity that we observed in Médina suggests that this species may exhibit opportunistic feeding behavior, depending on host availability. We acknowledge that the anthropophilic rate in Médina was unexpectedly low compared to the other sites. Although we did not conduct specific surveys to quantify the number of domestic animals in each locality, sociocultural differences may explain the observed variation. Future studies incorporating animal census data and finer-scale environmental variables would help clarify these patterns.

**Table 4. Number of genotypes and frequencies of *kdr* mutations in the VGSC gene of *Aedes aegypti* from six localities in Dakar.**

| Sites | Pheno-type | Kdr F1534C | | | Allele Freq | | Kdr V1016I | | | Allele Freq | Kdr V1016G | | | Allele Freq | | Kdr S989P | | | Allele Freq | |
|---|---|---|---|---|---|---|---|---|---|---|---|---|---|---|---|---|---|---|---|---|---|
| | | CC | FC | FF | | *p-value* | II | VI | VV | | GG | VG | VV | | *p-value* | PP | SP | SS | | *p-value* |
| | | | | | F(C) | | | | | F(I) | | | | F(G) | | | | | F(P) | |
| Médina | Alive | 0 | 40 | 54 | 0.21 | *0.8852* | 0 | 0 | 94 | – | 0 | 81 | 13 | 0.43 | *0.6638* | 3 | 83 | 8 | 0.47 | *0.477* |
| | Dead | 0 | 44 | 56 | 0.22 | | 0 | 0 | 100 | – | 0 | 89 | 11 | 0.44 | | 7 | 84 | 9 | 0.47 | |
| PE | Alive | 2 | 52 | 44 | 0.28 | *0.08164* | 0 | 0 | 99 | – | 0 | 95 | 4 | 0.47 | *0.6918* | 1 | 51 | 46 | 0.27 | *0.3247* |
| | Dead | 0 | 34 | 48 | 0.20 | | 0 | 0 | 81 | – | 0 | 79 | 2 | 0.48 | | 0 | 50 | 32 | 0.30 | |
| Mbao | Alive | 0 | 60 | 19 | 0.37 | *0.2093* | 0 | 0 | 79 | – | 0 | 79 | 0 | 0.50 | *1* | 0 | 79 | 0 | 0.50 | *1* |
| | Dead | 2 | 57 | 12 | 0.42 | | 0 | 0 | 71 | – | 0 | 71 | 0 | 0.50 | | 0 | 71 | 0 | 0.50 | |
| Ouakam | Alive | 1 | 7 | 69 | 0.02 | *0.4729* | 0 | 0 | 77 | – | 0 | 77 | 0 | 0.50 | *1* | 0 | 77 | 0 | 0.50 | *1* |
| | Dead | 0 | 6 | 84 | 0.08 | | 0 | 0 | 90 | – | 0 | 90 | 0 | 0.50 | | 0 | 90 | 0 | 0.50 | |
| Grand Yoff | Alive | 0 | 36 | 63 | 0.18 | *0.06558* | 0 | 0 | 99 | – | 0 | 99 | 0 | 0.50 | *1* | 0 | 99 | 0 | 0.50 | *1* |
| | Dead | 1 | 17 | 55 | 0.13 | | 0 | 0 | 73 | – | 0 | 73 | 0 | 0.50 | | 0 | 73 | 0 | 0.50 | |
| Guédiawaye | Alive | 0 | 0 | 85 | – | 1 | 0 | 0 | 85 | – | 0 | 79 | 6 | 0.46 | *0.004232* | 1 | 84 | 0 | 0.50 | *4.816e-12* |
| | Dead | 0 | 0 | 62 | – | | 0 | 0 | 62 | – | 0 | 47 | 15 | 0.37 | | 2 | 36 | 24 | 0.32 | |

CC, II, VV, and SS = homozygous wild-type; FC, VI, VG, and SP = heterozygous (mutant/wild-type); FF, II, GG, and PP = homozygous mutant. Freq = frequency of the mutant allele.

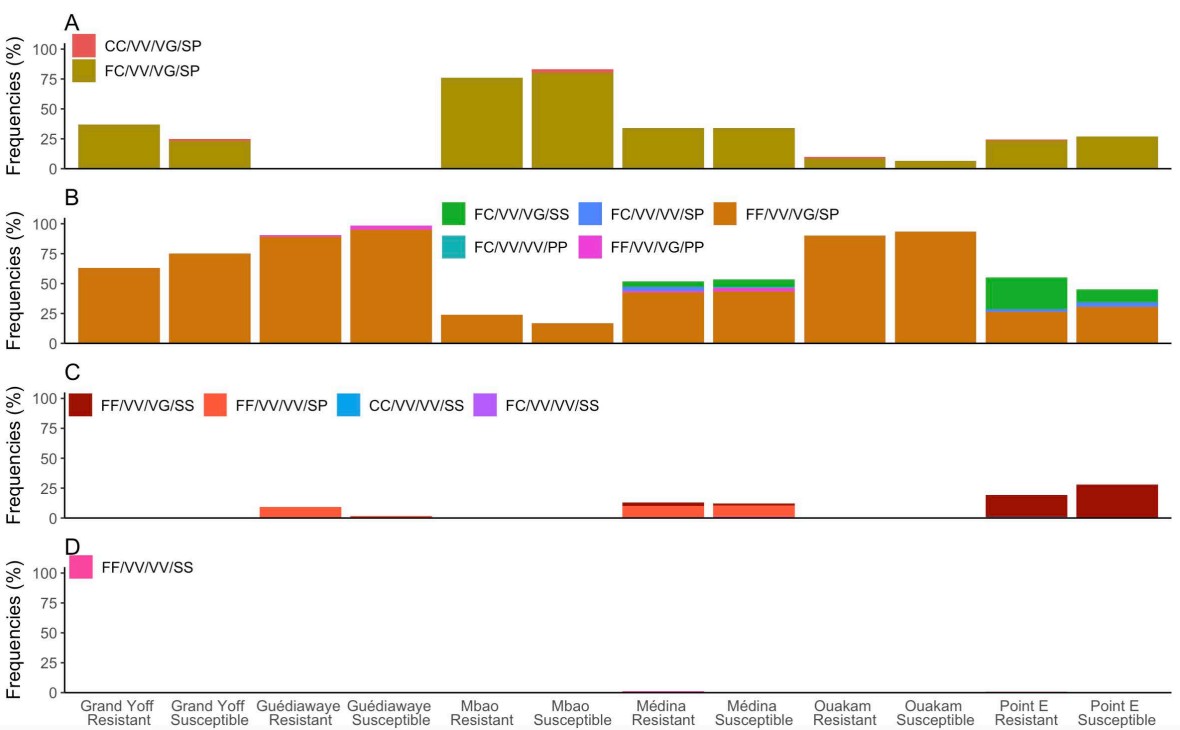

**Fig 9. Frequencies of four-locus genotypes for the voltage-gated sodium channel (VGSC) *kdr* mutations in phenotyped *Aedes aegypti*.** Each four-locus genotype is represented in the order of F1534C/ V1016G/ S989P. Genotypes are indicated as follows: F1534C – FF: wild type (susceptible), FC: heterozygote, CC: mutant (resistant); V1016G – VV: wild type (susceptible), VG: heterozygote; S989P – SS: wild type (susceptible), SP: heterozygote, PP: mutant (resistant). Freq indicates the frequency of each multilocus genotype in the tested mosquito population.

Furthermore, differences in resting and biting patterns of *Ae. aegypti* were observed across the six sampling sites. In Grand Yoff, Mbao, and Ouakam, females were primarily collected outdoors, suggesting more intense host-seeking activity in peri-domestic environments. In contrast, in Médina, Point E, and Guédiawaye, mosquitoes were collected both indoors and outdoors, indicating more flexible host-seeking. Host-seeking activity was predominantly diurnal in all sites, with 68.5% of females collected indoors and 56.5% outdoors during daytime, in agreement with findings from previous studies [46]. This suggested that mosquito control operations should be done mainly outdoor during the day. The Endophilic Rate (ER) also varied significantly among sites, with the highest value recorded at Point E. These spatial variations in behavioral patterns may be influenced by ecological and socio-environmental factors, such as housing type, human activity, or vector control practices, and underscore the need for locally tailored intervention strategies.

*Flavivirus* was detected in four of the 6 neighborhoods where adult mosquitoes were collected and tested (Grand Yoff, Guédiawaye, Mbao, and Point E). The detection of virus-positive pools in multiple, geographically dispersed sites indicates a possibly widespread viral circulation in the urban environment, highlighting the risk of human transmission across Dakar. Consequently, implementing regular control strategies to minimize human/vector contact is highly recommended for limiting arbovirus transmission in the Dakar area. Phylogenetic analysis of DENV-3 strains isolated during monitoring revealed their classification within genotype III. The serotype 3 detected in this study was responsible for the first dengue outbreak recorded in Dakar in 2009 [47] and is currently the most frequency associate outbreaks in Senegal [48]. The high risk of transmission in Senegal may be due to the geographical proximity to Benin and Burkina Faso, coupled with significant human migratory flows. Additionally, rapid urbanization has been identified as a key factor facilitating dengue transmission [49]. Attempts to sequence the viral RNA from the two other pools were unsuccessful, probably due to low RNA quality or quantity. Mosquito-based viral surveillance is a fundamental component of arbovirus control, as it helps in detecting and monitoring local viral activity, assessing endemicity levels, and identifying high-risk areas [50].

Assessing the susceptibility of *Ae. aegypti* to different insecticides under laboratory conditions is an important indicator for evaluating the effectiveness of insecticides-based control strategies. In this study, all tested *Ae. aegypti* populations showed resistance to pyrethroids when assessed using the new diagnostic doses recommended by the WHO for *Aedes* mosquitoes, despite a high knockdown rate. Some populations also exhibited suspected resistance at intensity doses (5 or 10 times higher than diagnostic doses). The presence of pyrethroid resistance in *Ae. aegypti* had previously been demonstrated in Senegal using diagnostic doses developed for *Anopheles* [7]. The widespread resistance observed in *Ae. aegypti* populations in Dakar to all insecticides tested including carbamates, organophosphates, and organochlorines may be explained by multiple sources of insecticide exposure. With the contact man-vector, insecticides are commonly used in households for nuisance mosquito control. In particular, pyrethroids such as permethrin, deltamethrin, and allethrin are widely used in Dakar households for nuisance mosquito control in the form of sprays, coils, and aerosols [6]. Carbamates such as propoxur are used in reactive intervention campaigns during epidemics, and resistance to this class of insecticides (bendiocarb) has been observed [7]. Additionally, among our six sites, Guédiawaye and Mbao are part of the Niayes zone. This area is, characterized by intensive urban agriculture, particularly market gardening, which involves frequent use of organophosphates and pyrethroids [51]. *Ae. aegypti* is characterized by a limited flight range, generally not exceeding 100–200 meters under natural conditions [52], although some individuals have been reported to travel up to 800 meters [53]. This could contribute to a constant environmental pressure selecting for resistant individuals. Similar patterns have been observed in Ghana, where extensive use of agricultural insecticides has been linked to increased resistance levels in mosquito populations [54,55]. Although pyrethroid-treated bed nets and IRS are primarily aimed at malaria vectors, their widespread and long-term use in Senegal may also exert indirect selection pressure on *Ae. aegypti* populations exposed to these interventions in peri-domestic environments [56]. Given these findings, regular monitoring of insecticide resistance in *Ae. aegypti* populations is essential to inform vector control strategies. It allows timely adjustments in the choice of insecticides, the rotation of active ingredients to manage resistance, and the adoption of complementary measures such as larvicide or

the introduction of Wolbachia infected mosquitoes. This evidence-based approach helps optimize public health interventions and maintain the effectiveness of vector control programs.

This study also provides the first documented evidence of *kdr* mutations in *Ae. aegypti* populations in Senegal. Specifically, the homozygous resistant of 1534C and 989P mutations (specifically CC an PP) and the heterozygote resistant of 1016G (VG) were detected at in both resistant and susceptible populations. Except for the V1016G and S989P mutations in Guédiawaye, no significant differences in *Ae. aegypti* allele frequencies were observed between resistant and susceptible mosquitoes despite the resistance to pyrethroids. A prior study conducted in Senegal in 2019 reported a complete absence of *kdr* mutations [7]. The recent detection of *kdr* mutations in *Ae. aegypti* populations in Senegal, including S989P, V1016G, and F1534C, suggests a possible emergence or introduction of resistance alleles that may expand over time. This pattern is similar to studies from Burkina Faso, where the V1016I mutation was first reported at moderate frequencies and progressively increased over the years [57]. The potential for a rapid increase of resistance alleles once they emerge in a population, particularly under continuous insecticide pressure, is highlighted by this trend. Monitoring these dynamics is therefore critical for anticipating resistance spread and adapting vector control strategies accordingly [58]. In addition, previous studies conducted in Senegal concluded that metabolic resistance by detoxifying enzymes is the main mechanism leading to pyrethroid resistance in *Ae. aegypti*. In particular, high expression levels of cytochrome P450 genes such as CYP9J26, CYP9J28, CYP9J32, CYP6BB2, CYP9J32 and CYP9J26 were noted. A significant overexpression of the glutathione S-transferase gene GSTD4 and the esterase gene CCEae3a was observed on all tested *Ae. aegypti* in Senegal [7]. Several multi-locus genotypes combining these mutations were detected at comparable frequencies in both resistant and susceptible mosquitoes. These findings highlight the complexity of resistance in Senegal and support the need for comprehensive surveillance integrating both target-site and metabolic mechanisms. Although the triple mutant homozygote (S989P+V1016G+F1534C) has not yet been detected in Senegal, its potential emergence remains a concern, as Hirata et al. (2014) [59] demonstrated that this combination can significantly increase pyrethroid resistance and potentially compromise vector control strategies.

This suggests that their involvement in pyrethroid resistance cannot be confirmed. One limitation of this study is that the V410L *kdr* mutation, which has been increasingly reported in *Ae. aegypti* populations across West Africa, was not included in our screening panel [11,14,15]. Given its potential role in pyrethroid resistance, future investigations should consider incorporating the V410L mutation in molecular monitoring efforts in Senegal.

## Conclusion

Our results demonstrate that the implementation of a monitoring system using ovitraps is both sensitive and efficient, requiring minimal resources for routine inclusion in dengue control programs. Moreover, a limited number sites is sufficient, as no significant differences were observed in the temporal dynamics of eggs among neighborhoods.

Our results also provide the first data on the distribution, breeding sites, blood-feeding behavior, biting patterns, and resting site preferences for adult *Ae. aegypti* in Dakar. Our findings indicate that, *Ae. aegypti* is well-established in Dakar. The effectiveness of vector control measures may be compromised by the exophagic and diurnal behavior of this species.

The detection of kdr mutations F1534C, V1016G, V1016I, S989P, along with the overexpression of several detoxification genes is alarming. It is important to assess the intensity of resistance and explore complementary control strategies (such as *Bacillus thuringiensis var. israelensis* (Bti), *Wolbachia* infection, sterile insect techniques and genetic manipulation, particularly in African countries where dengue epidemics can have severe health impacts.

## Supporting information

**S1 Table. Number of adult mosquitoes collected from August 2022 to July 2023 in Dakar.**
(DOCX)

**S2 Table. Mean number of mosquitoes species collected per localities per month in Dakar from August 2022-July2023.**
(DOCX)

**S3 Table. Knockdown times (KDT$_{50}$ and KDT$_{95}$, with 95% confidence intervals) of _Aedes aegypti_ following exposure to different concentrations of pyrethroids across six localities in Dakar/ August 2022 to July 2023.**
(DOCX)

**S4 Table. Number of genotypes and frequencies of _kdr_ mutations in the VGSC gene of _Aedes aegypti_ from six localities in Dakar.**
(DOCX)

## Acknowledgments

We thank all the study participants, the field workers and the lab technicians, PhD Research Associate in Institut Pasteur de Dakar. **Authors' Disclaimer Statement:** The views expressed in this article are those of the authors and do not necessarily reflect the official policy or position of the U.S. Department of the Navy, U.S. Department of Defense, nor the U.S. Government. Opinions, interpretations, conclusions, and recommendations are those of the authors and are not necessarily endorsed by the U.S. Navy. **Copyright Assignment statement:** CDR Harwood and Dr. Nimo-Paintsil are military service members or employees of the U.S. Government. This work was prepared as part of their official duties. Title 17 U.S.C. §105 provides that 'Copyright protection under this title is not available for any work of the United States Government.' Title 17 U.S.C. §101 defines a U.S. Government work as a work prepared by a military service member or employee of the U.S. Government as part of that person's official duties.

## Author contributions

**Conceptualization:** Ndeye Marie Sene, Shirley Nimo-Paintsil, Diawo Diallo, Ibrahima Dia, Scott C. Weaver, Samuel Dadzie, James F Harwood, Mawlouth Diallo.

**Formal analysis:** Ndeye Marie Sene, El Hadji Malick Ngom, Babacar Diouf, Moussa Moise Diagne.

**Funding acquisition:** Shirley Nimo-Paintsil, Samuel Dadzie, James F Harwood, Mawlouth Diallo.

**Investigation:** Ndeye Marie Sene, Moussa Gaye, El Hadj Ndiaye, El Hadji Malick Ngom, Babacar Diouf, Faty Amadou Sy.

**Methodology:** Ndeye Marie Sene, Moussa Gaye, El Hadj Ndiaye, Faty Amadou Sy, Moussa Moise Diagne, Alioune Gaye, Diawo Diallo.

**Project administration:** Shirley Nimo-Paintsil, Scott C. Weaver, Samuel Dadzie, James F Harwood, Mawlouth Diallo.

**Resources:** Shirley Nimo-Paintsil, Scott C. Weaver, Samuel Dadzie, James F Harwood, Mawlouth Diallo.

**Software:** Ndeye Marie Sene, Moussa Gaye, Alioune Gaye, Diawo Diallo, Ibrahima Dia.

**Supervision:** Shirley Nimo-Paintsil, Alioune Gaye, Diawo Diallo, Ibrahima Dia, Scott C. Weaver, Samuel Dadzie, James F Harwood, Mawlouth Diallo.

**Validation:** Ndeye Marie Sene, Shirley Nimo-Paintsil, Mawlouth Diallo.

**Visualization:** Ndeye Marie Sene.

**Writing – original draft:** Ndeye Marie Sene, Moussa Gaye.

**Writing – review & editing:** Ndeye Marie Sene, Shirley Nimo-Paintsil, Moussa Gaye, El Hadj Ndiaye, El Hadji Malick Ngom, Babacar Diouf, Faty Amadou Sy, Moussa Moise Diagne, Alioune Gaye, Diawo Diallo, Ibrahima Dia, Scott C. Weaver, Samuel Dadzie, James F Harwood, Mawlouth Diallo.

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
