## [Decision Letter · Decision Letter 0]

27 May 2025

Entomological surveys and the evolution of insecticide resistance in Aedes aegypti populations in Dakar, the capital city of Senegal: first detection of kdr mutation

Dear Dr. Sene,

Thank you for submitting your manuscript to PLOS Neglected Tropical Diseases. After careful consideration, we feel that it has merit but does not fully meet PLOS Neglected Tropical Diseases's publication criteria as it currently stands. Therefore, we invite you to submit a revised version of the manuscript that addresses the points raised during the review process.

Please submit your revised manuscript within 60 days Jul 26 2025 11:59PM. If you will need more time than this to complete your revisions, please reply to this message or contact the journal office at plosntds@plos.org. Please include the following items when submitting your revised manuscript:

We look forward to receiving your revised manuscript.

Kind regards,

Clarence Mang'era, PhD

Academic Editor

Paul Mireji

Section Editor

Shaden Kamhawi

co-Editor-in-Chief

Paul Brindley

co-Editor-in-Chief

**Additional Editor Comments:**

While your study addresses an important public health issue in Senegal with potential global impact, the manuscript requires extensive amendments in several key areas such as statistical analysis, ethical approval documentation and well supported conclusions to meet journal standards.

**Journal Requirements:**

At this stage, the following Authors/Authors require contributions: Ndeye Marie Sene, Shirley Nimo-Paintsil, Moussa Gaye, El Hadj Ndiaye, El Hadji Malick Ngom, Babacar Diouf, Faty Amadou Sy, Moussa Moise Diagne, Alioune Gaye, Diawo Diallo, Ibrahima Dia, Scott C. Weaver, Samuel Dadzie, James F Harwood, and Mawlouth Diallo. Please ensure that the full contributions of each author are acknowledged in the "Add/Edit/Remove Authors" section of our submission form.

- TM on page: 44.

Potential Copyright Issues:

- Figures 1 and 7. Please (a) provide a direct link to the base layer of the map (i.e., the country or region border shape) and ensure this is also included in the figure legend; and (b) provide a link to the terms of use / license information for the base layer image or shapefile. We cannot publish proprietary or copyrighted maps (e.g. Google Maps, Mapquest) and the terms of use for your map base layer must be compatible with our CC BY 4.0 license.

6) We note that your Data Availability Statement is currently as follows: "All relevant data are within the manuscript and its Supporting Information files". Please confirm at this time whether or not your submission contains all raw data required to replicate the results of your study. Authors must share the “minimal data set” for their submission. PLOS defines the minimal data set to consist of the data required to replicate all study findings reported in the article, as well as related metadata and methods (https://journals.plos.org/plosone/s/data-availability#loc-minimal-data-set-definition).

- The points extracted from images for analysis..

7) Please ensure that the funders and grant numbers match between the Financial Disclosure field and the Funding Information tab in your submission form. Note that the funders must be provided in the same order in both places as well.

**Reviewers' Comments:**

Reviewer's Responses to Questions

**Key Review Criteria Required for Acceptance?**

**Methods**

-Are the objectives of the study clearly articulated with a clear testable hypothesis stated?

-Is the study design appropriate to address the stated objectives?

-Is the population clearly described and appropriate for the hypothesis being tested?

-Is the sample size sufficient to ensure adequate power to address the hypothesis being tested?

-Were correct statistical analysis used to support conclusions?

-Are there concerns about ethical or regulatory requirements being met?

Reviewer #1: Methods have many omissions making it difficult to assess robustness in several places. Statistical analyses limited or unclear and do not appear to support conclusions properly. Stated that no ethical review required

see main review for further comments

Reviewer #3: Are the objectives of the study clearly articulated with a clear testable hypothesis stated?

The objective of the study are clearly articulated.

-Is the study design appropriate to address the stated objectives?

The study design is appropriate to address the objective.

-Is the population clearly described and appropriate for the hypothesis being tested?

The population is clearly described and appropriate for the hypothesis being tested.

-Is the sample size sufficient to ensure adequate power to address the hypothesis being tested?

The sample size is sufficient.

-Were correct statistical analysis used to support conclusions?

The statistical

-Are there concerns about ethical or regulatory requirements being met?

The ethical approval is required.

**Results**

-Does the analysis presented match the analysis plan?

-Are the results clearly and completely presented?

-Are the figures (Tables, Images) of sufficient quality for clarity?

Reviewer #1: Results presentation generally ok though some amendments required and not all should be in main text or require modification

Reviewer #3: -Does the analysis presented match the analysis plan?

The analysis plan is good.

-Are the results clearly and completely presented?

The results are clear. There are sections that the author has mixed the methodogly and results. Section starting line 159 - 176

-Are the figures (Tables, Images) of sufficient quality for clarity?

The figures are of sufficient quality.

**Conclusions**

-Are the conclusions supported by the data presented?

-Are the limitations of analysis clearly described?

-Do the authors discuss how these data can be helpful to advance our understanding of the topic under study?

-Is public health relevance addressed?

Reviewer #1: In my opinion some key conclusions are not well supported

No limitations are noted - but should be

Public health relevance needs clarification

Reviewer #3: -Are the conclusions supported by the data presented?

Yes

-Are the limitations of analysis clearly described?

No

-Do the authors discuss how these data can be helpful to advance our understanding of the topic under study?

Yes

-Is public health relevance addressed?

Yes

**Editorial and Data Presentation Modifications?**

Reviewer #1: No very major changes are required

Reviewer #3: Entomological surveys and the evolution of insecticide resistance in Aedes aegypti

populations in Dakar, the capital city of Senegal: first detection of kdr mutation

Line 58 -68 Introduction section: There is a disconnect between the title and the manner in which the introduction starts. The authors start by describing dengue (one of the many viruses transmitted by Aedes disconnecting the reader from the title. It would be ideal if the authors started this section by describing the vector (as it is the primary focus of the study. This can then be followed by descriptions of infections and disease burdens associated with the vectors, challenges in control and description of the resistance mechanisms

Line 99- 102 – The use of etc should be avoided. Susceptibility is mis pelt

110 -Ethic statement – as a clarification shouldn’t a protocol undergo some form of scientific and ethical approval process prior to implementation?

Line 149- 150 – “Sentinel traps baited with a chemical Lure and CO2”. This is grammatically wrong.

Line 153 – “Resting mosquitoes were sampling using a Prokopack aspirator”. Correct this

Line 159 – 169 - the authors describe both methods and results (number of pools generated in this section and need to address grammatical errors for clarity. The same applies to lines 188-189 where the authors preempt the results in the methodology section

Line 256 table 1 Ovitrap indices– is there a reason why fewer ovitraps were placed in Ngor and Pikine and how did this influence the comparison between the sites

Line 153: Resting mosquitoes were "sampled" using a..........

Figure 6: Line no. 320-321 - The author should rephrase this sentence for clarity. "Phylogenetic relationship of newly generated DENV whole genomes from various serotypes and genotypes available in GenBank sequences and those previously available."

**Summary and General Comments**

Reviewer #1: General comments

This paper contains useful data on entomological surveillance, phenotypic insecticide resistance, detection of arbovirus in mosquitoes and perhaps most significantly the detection of kdr mutations. However, the data are poorly analysed and without this conclusions are not well supported. Methods as presented are missing a lot of important details, precluding assessment of what was actually done and its robustness. The Discussion has many erroneous or misleading statements including misreporting of results from this and other studies, and apparent lack of understanding of how unusual the reported discoveries of the kdr mutations, which include Asian variants, are. Overall very substantial revision in all sections is required to bring this work to a publication standard.

Specific comments

Abstract

The study aim is stated as assessing risk of arbovirus transmission but there is nothing further in the abstract about how this study relates to this aim

Clarify that sampling was for one full year spanning 2022-2023; just giving the years could be read as meaning there were two full sampling years

Clarify that the total was 150 houses distributed among 15 neighbourhoods (i.e. 10 per neighbourhood, rather than 150 each)

It would be helpful to give an indication of insecticides tested and to which resistance detected

Some indication of conclusions or study significance would be useful in the abstract – some are present in the author summary but not currently in the abstract

Introduction

Line 79-80 could you give examples of the control methods used and when/where applied in Senegal, e.g. are they applied at specific outbreak times and locations? This would give helpful context.

Line 81-82 I am aware of two studies though the 2012 work was limited but you may wish to cite anyway since you say studies (plural) here. Also, would be helpful to give more detail here on what resistance was detected and to which types of insecticide, rather than saying reduced susceptibility which could be interpreted in different ways

Line 90 other mutations (V1016I and 410L) have also been detected in at least some of these neighbours. This is an important point to give context to your subsequent findings. Has the V410L every been looked for in Senegal?

Lines 91-92. I think it is worth noting here that the lack of detection comes from genotyping of a large number of samples in the cited study. This is important to emphasise that limited surveillance effort is a less likely explanation than low frequency

Line 93-94 I know this kind of statement is frequently made in publications, but to say it is essential for outbreak control given that pyrethroid resistance appears to be known already and is likely a suboptimal control solution seems an overstatement. Some context here is required to justify why this is important (e.g. the capacity of different kdr mutations to combine and elevate resistance or geographical variability in resistance (with respect to results in ref 7) that could mean that pyrethroids might be useful in some places, but spatiotemporal surveillance required and kdr mutation typing might help facilitate this).

Line 94, this is not true – coexistence of kdr mutations with metabolic resistance is well known especially in SE Asia – see e.g. review papers on Aedes aegypti resistance.

Line 102 suggests that such data are limited but not absent. Please cite relevant reference(s) or change wording if genuinely absent

Line 105 in the context of the way the study was performed here, what data would show a difference between host-seeking, resting and feeding behaviours? This needs some additional detail to specify assumptions with respect to the interpretation of the different collection methods.

Materials and Methods

Ethics: in many countries (including in Africa) this would now require ethical approval because it involves collections from households, which is regarded as human involvement. So I think it is essential that you clarify your statement here to say that this did not require ethical approval in Senegal (or Dakar, whichever is the case) to avoid giving the impression that this would be the case for this kind of work elsewhere.

Line 142 ‘lined with a wooden stick’ I think should read ‘included a wooden stick’

Line 147-148 why were ovitrap collections performed in 15 sites and adult collections in 6 - what was the rationale for this subset? Also, you should make it clear which district each site is in so that reader does not have to work this out from the figure or earlier text.

Line 150 was this a BG lure? If so, which one was used and how often were these replaced?

Line 150-151 how many houses per site and where was the outdoor trap located?

Line 155 where were outdoor collections made?

Line 156 what is the rationale for the pooling strategies (i.e. N=) and what is the purpose of the ‘other species’ pools given that these were not used for ABV detection?

Line 155-160 it is not stated how mosquitoes were killed, how quickly they were preserved nor what the preservation method was; for subsequent RNA extractions these are important details. Presumably also these were whole mosquitoes, rather than head/thorax or salivary gland dissections – add this for clarity.

Line 160 which arboviruses?

Line 163 and line was = were

Line 163-4 32 x 20 doesn’t equal the stated total of 613

Line 166 above it states that super-pools consisted of 20 pools but here says that consisted of 6-7 pools

Line 165 why were males included in the screening? I assume fed and unfed were treated separately to detect infected (maybe infectious) vs infectious females. Please clarify the rationale for the different groups used.

Line 173 dengue is not capitalised

Line 174 For virus detection by RT-PCR what thermal cycler was used and more importantly what threshold ct or cq value, and what were there positive controls (what strains and supplier)?

Line 175 flavivirus and alphavirus capitalised here but not in line 170

Line 182 is a reference website available for these recommendations?

Line 189 please give accession numbers

Line 175-176 It isn’t clear here or from anything preceding in the manuscript why this sequencing is being done. From reading the qPCR paper cited I think it is because the method just detects flaviviruses, unless it has been modified here. If this is correct, why was the alphavirus chikungunya used as a positive and why were subsequent pools tested for flavivirus and alphavirus. I’m not familiar with the Twist research panel and I suspect this will be the case for many readers. The link for reference 19 is dead but looking at the website the details are limited. More information is required here and citations to papers if available rather than just a website link. The method might be great, but I can’t assess this from the information I could find.

Line 193-207 please explain the rationale for the doses used. I can understand the lower doses which are Aedes aegypti standards now (and it is good that these are used) but the higher ones look like from 5X Anopheles intensity assays. I’m assuming the reason for this was based on paper availability, but this needs to be explained that these were being used as a way of looking at resistance intensity using available (rather than optimal) papers.

Line 194-195. In the abstract it states that females for testing were reared from collected eggs but this detail is omitted here. Did these come from ovitraps or collected blood fed females laying eggs? In either case, how many eggs were reared from each ovitrap or female’s brood i.e. how was it ensured that the tested cohorts were not composed of many siblings, which could impact the results? Also were all returned to the same laboratory for testing?

Line 201 should chlorpyrifos ethyl read methyl?

Line 204 when was knockdown recorded – at 1h?

Line 213-214 I am not necessarily questioning the validity of this but given the expectation of very low frequencies, leading to a lack of past detection, why were both resistant and susceptible chosen for genotyping (rather than just resistant, at least as an initial subset to determine whether mutations were present). Also why the sample was divided among the four pyrethroids – again, some explanation of rationale would be helpful

Line 212 This is an unusual choice for screening, given the previous lack of detection of the Asian kdr mutants (989P and 1016G) in West Africa, but the previous detection of 410L which as a standalone may be the most important kdr mutation in Aedes aegypti. Please provide a rationale for the choice.

Line 227 assuming ANOVA means the standard method based on the normal distribution, was this checked for?; please give details

Line 235 Please explain from what data these KDT values were calculated, Did you record KD data at 10 minutes intervals? If so, please clarify this in the description of KD recording data.

Results

Lines 244-45 Please provide definition of epidemiological weeks

Lines 247-53 this analysis compares variation between sites but does not incorporate temporal effects – I would suggest to instead use a GLM with time of collection (could be grouped to month or season if desired) and the location x time interaction to evaluate consistency of variation over time

Also, and especially given the huge amount of work quantifying the EDI, and that EDI data are presented in figure 2, some similar analysis here would be warranted

Table 1 – how do stars indicate significance – i.e. with respect to what? All seem significant except Guediawaye but the ODI for this site does not seem extreme. Is there some error here?

Table 2 is slightly confusing because in the text you refer to ‘remaining species’ but in the table to ‘other species’ and do not list those which are included in this category, even though in some cases frequencies are similar to Aedes and you have included them in the text. Given that the focus is on Aedes, why not just include Aedes in the table to make it more easily readable? It is almost always the case that Cx quinq will be the most common in collections in Africa but they are not subject to any an analysis so maybe just present the full data for all species as a supplement – all seem to have been identified, representing considerable effort so it seems a waste for these data not to be presented somewhere.

Line 274-280 I think something is missing in the description of statistics used because chi-square would only identify an overall difference (maybe relates to strange significance classifications in table 1 also); was some post hoc test used as well?

Line 281-284 figure 3 represents pooled data without associated statical support for the statement that all sites showed same peak of Aedes abundance – see comment above about statistical testing, or at least provide confidence limits

Line 288 - 303 statistical support required to support assertion of locality differences in indoor vs outdoor resting and host preference. Also, no detail of methodology for blood meal typing is provided in the Methods (I assume is Kent and Norris given the host species identified). The HBI for Medina is also unusually low. Some description of the adult sample sites and any variation among them would be useful in the Methods to help determine (if significantly lower) why this might be the case.

Lines 306-7 pool size does not appear to correspond that stated in Methods which for Aedes aegypti was stated as 20 per pool – for females this multiplier is close, but not exactly correct; for males it is not close

Line 308 which pools (male, female-fed, female-unfed) were positive? Also clarify how detected – RT-PCR or sequencing

Line 310-311 what does successfully serotyped as dengue 3 mean? Does this mean that the other two were some other serotype or not dengue? Clarity in what the RT-PCR method detects as positive may answer this question (see comment above). Also it seems strange that sequencing, which I would imagine would be highly sensitive, did not detect something in the positive pools. However, this is difficult to evaluate because of the lack of detail presented on the methods used.

Figure 6 legend. What does ‘Root-to-tip regression and time-scaled phylogenetic analysis show linear evolution between newly generated Dengue virus sequences obtained in Dakar (August-2022 to July-2023) and the other West African genotype sequences’ mean? Where is the time-scaling?

Line 333 please clarify what the actual intensity assays represented in terms of X – since these appear to be Anopheles higher dose papers but Aedes 1X papers they will not necessarily correspond to 5X and 10X

Fig 7 what does the size of the dot represent – is it mortality? Also the resistance threshold terminology needs aligning to that used in the text

Fig 8 which pyrethroid? Also the only text referring to this is the single sentence in 346-347 which refers to the KDT values but these are not shown, and the statement is obvious – that the values would be higher for diagnostic than higher doses. Unless there is some further analysis here, this figure is not worth inclusion, at least in the main text.

Line 367-368 how was this tested – on alleles or genotypes and what test was used? In the table the exact P-values should also be provided rather than P>0.1 for everything. It also appears that all sites and insecticides were tested separately but in the text it refers to pooled insecticides. The table is too large for inclusion in the main text and I would suggest to move to the supplement and provide here a reduced version which just shows the total of dead and alive with study sites pooled. Further investigation into associations is also required because the sample sizes are rather small to give well powered tests when subdivided as they are here.

Fig 9 is difficult to read and the order of loci stated in the legend is reversed with respect to thye axis labels. Maybe the genotypes could be ordered according to the number of resistant alleles present in the four locus genotype, and the resistant alleles indicated in a different colour in the axis labels to improve interpretability? It is also well demonstrated that the combination and number of resistant alleles across kdr loci is important for resistant – more so than single locus genotypes. Statistical testing of resistance association of the multilocus genotypes is required.

Discussion

Lines 415-418 I think some statistical support for variation in species diversity is required in the Results

Line 433 this sentence needs some context – why detection in four (of many) pools suggests substantial viral circulation

Line 440 migration of humans or mosquitoes?

Line 452 also agriculture

Line 455-457 it would be helpful here to give some more detail of how the monitoring can inform interventions

Line 459 this does not match the results reported – frequencies are not low and line 464 1016G is not absent (also at quite high frequency)

Lines 465-466 This is incorrect. None of these cited studies detected 989P and 1016G so the frequencies in Dakar were not lower. For two of the studies they definitely did not genotype for these mutants, and for the third, for which I could not access the full text, I would be surprised if they did but did not report absence. They did all screen for V410L though, which is a significant omission from this study

Line 467-470. This is wrong – these mutations can cause very high level resistance when in combination (see reviews or original Hirata paper). I guess what is referred to here is that association was not detected here but as noted in comments above, the statistical testing procedures need revisiting.

Line 470 similar low frequencies of what? This study also did not screen for S989P or V1016G

Line 471 study 7 showed overexpression of these genes but not resistance association. This was inferred indirectly from the lack of kdr mutations found and the presence of phenotypic resistance. Therefore this conclusion is not robust, especially now that kdr mutations have been detected. Lack of contribution of kdr mutations to resistance when multiple kdr alleles are detected would also be very surprising given the functional validation of these and multiple studies showing a strong contribution to resistance.

Line 493 1016I was not detected and detox genes were not examined in this study and whether they would have been overexpressed as they were in study 7 is speculation, especially given that the populations appear to have changed substantially with respect to kdr.

Reviewer #2: I have read with great interest this manuscript profiling the resistance of Aedes aegypti samples from Dakar to insecticides and assessing the bionomics of this species. However, some clarifications are needed to consider this manuscript for publication in PNTD.

Line 20 “Aedes. Aegypti”.

At the beginning of the sentence, you must write Aedes in full. Please correct it across the manuscript.

Lines 79-80. “Current emergency control measures rely primarily on insecticide-based strategies, particularly, pyrethroid adulticides”. There is an any reference supporting this?

Lines 94-97. “The coexistence of kdr mutations with an overexpression of detoxification enzymes, a phenomenon never observed elsewhere, could greatly undermine the effectiveness of pyrethroid insecticides, which currently constitute the primary vector control method”. I’m not sure that this is true. Please see Ishak et al., 2015; Ishak et al. 2017 https://pubmed.ncbi.nlm.nih.gov/25888775/;
https://journals.plos.org/plosntds/article?id=10.1371/journal.pntd.0005302

Ethic statement: The justification provides to explain the absence of ethical clearance is not clear enough since samples were collected in private settings.

Line 175. “Flavivirus” “Alphavirus” should be italicized.

Line 193. Authors collected samples or populations. This need to be clarified across the manuscript.

Line 209-214. These sentences need to be reformulated. Mosquitoes genotyped were dead and alive after exposure to insecticide. Please indicated the minimum number of dead and alive specimens analysed in each sample.

Line 264-266. “Aedes.”, “Mansonia.” And “Culex.”. Please remove the dot after the name of genus.

Line 266. How can the authors be sure this is An. gambiae from morphology?

Aspiration was done indoor and outdoor. It would be important to indicate the place outdoor.

I suggest also to test whether the number of Aedes collected outdoor vs indoor is significant different or not?

In the manuscript, the authors state that mosquitoes were collected at 7am and 7pm using the BG Sentinel trap. I have serious doubts that the Aedes mosquitoes considered to have been collected at night were really collected at night. I believe they were collected in the early morning between 5am and 7am.

Which lab susceptible reference strain was used? How much could the authors be confident for the quality of impregnated papers used?

In fig 7. What does tolerant mean?

What could explain the resistance of Ae. aegypti samples from Dakar to all insecticides tested including carbamates, organophosphates, and organochlorines? These composed are currently used in Dakar for Aedes or mosquitoes’ control? This need to be discussed in the manuscript.

Differences found in resting and biting patterns between sampling sites should be discussed.

Reviewer #3: This study provides a comprehensive assessment of the spatial distribution, seasonal dynamics, and insecticide resistance mechanisms of Aedes aegypti in Dakar. Key findings include the widespread presence of Ae. aegypti across all surveyed sites, with increased activity observed between August and October, coinciding with the rainy season and heightened risk of arbovirus transmission. The detection of dengue virus (DENV) in mosquitoes collected from September to December highlights ongoing viral circulation and potential outbreak risk during this period.

The investigation into insecticide resistance revealed that mosquito populations exhibited resistance to WHO-recommended diagnostic doses, which could compromise current control efforts. Although the identified kdr mutations (F1534C, V1016I/G, and S989P) were present at low frequencies, their existence suggests emerging resistance mechanisms that may intensify over time.

**Figure resubmission:**

**Reproducibility:**



---

## [Decision Letter · Decision Letter 1]

11 Sep 2025

Please submit your revised manuscript within 30 days Oct 11 2025 11:59PM. If you will need more time than this to complete your revisions, please reply to this message or contact the journal office at plosntds@plos.org. Please include the following items when submitting your revised manuscript:

* A rebuttal letter that responds to each point raised by the editor and reviewer(s). You should upload this letter as a separate file labeled 'Response to Reviewers '. This file does not need to include responses to any formatting updates and technical items listed in the 'Journal Requirements' section below.

Revised Manuscript with Track Changes
Manuscript

Clarence Mang'era, PhD

Academic Editor

Shaden Kamhawi

co-Editor-in-Chief

Paul Brindley

co-Editor-in-Chief

**Additional Editor Comments :**

Line 31. Kdr mutations should be fully written for the first time.

Line 33: Please use the lowercase "dengue."

Line 41: It would be helpful to briefly list the specific multiple resistance mechanisms that were detected here.

Lines 121-124. The coexistence of kdr mutations and metabolic resistance mechanisms has also been reported in sub-Saharan Africa (Montgomery et al., 2022)

Lines 130-131. The objective of this specific part of the study could be stated more explicitly for the reader.

Line 194. CO2 include a subscript

Line 216-217. “The mosquitoes were homogenized and used for RNA extraction and Blood Feeding Patterns”. This sentence is not clear.

Please clarify the quality/standard of the insecticide-impregnated papers used, including the justification of concentration values used.

The legend of Figure 3 is incomplete.

Table 4. The legend is incomplete. CC, PP, II, CF, FF, VI, Freq ….

Line 592-592. Could you specify which of the insecticides listed in line 591 are commonly used for household nuisance control in Dakar? A reference for this common usage would be ideal here.

For context, please include the approximate distance between your study sites in Dakar and the Niayes zone. Also, a brief note on the known dispersion capacity of Aedes mosquitoes would strengthen the discussion.

**Reviewers' comments:**

**Key Review Criteria Required for Acceptance?**

**Methods**

-Are the objectives of the study clearly articulated with a clear testable hypothesis stated?

-Is the study design appropriate to address the stated objectives?

-Is the population clearly described and appropriate for the hypothesis being tested?

-Is the sample size sufficient to ensure adequate power to address the hypothesis being tested?

-Were correct statistical analysis used to support conclusions?

-Are there concerns about ethical or regulatory requirements being met?

Reviewer #2: (No Response)

**Results**

-Does the analysis presented match the analysis plan?

-Are the results clearly and completely presented?

-Are the figures (Tables, Images) of sufficient quality for clarity?

Reviewer #2: (No Response)

**Conclusions**

-Are the conclusions supported by the data presented?

-Are the limitations of analysis clearly described?

-Do the authors discuss how these data can be helpful to advance our understanding of the topic under study?

-Is public health relevance addressed?

Reviewer #2: (No Response)

**Editorial and Data Presentation Modifications?**

Reviewer #2: (No Response)

**Summary and General Comments**

Reviewer #2: (No Response)

PLOS authors have the option to publish the peer review history of their article (what does this mean? ). If published, this will include your full peer review and any attached files.

**Do you want your identity to be public for this peer review?** For information about this choice, including consent withdrawal, please see our Privacy Policy .

Reviewer #2: No

**Figure resubmission:**

**Reproducibility:**

To enhance the reproducibility of your results, we recommend that authors of applicable studies deposit laboratory protocols in protocols.io, where a protocol can be assigned its own identifier (DOI) such that it can be cited independently in the future. Additionally, PLOS ONE offers an option to publish peer-reviewed clinical study protocols. Read more information on sharing protocols at https://plos.org/protocols?utm_medium=editorial-email&utm_source=authorletters&utm_campaign=protocols

---

## [Editor Report · Decision Letter 2]

17 Oct 2025

Dear Dr. Sene,

We are pleased to inform you that your manuscript 'Entomological Surveys and Insecticide Resistance in the Dengue Vector Aedes aegypti in Dakar, Senegal: First Detection of the kdr Mutation' has been provisionally accepted for publication in PLOS Neglected Tropical Diseases.

Best regards,

Clarence Mang'era, PhD

Academic Editor

Paul Mireji

Section Editor

Shaden Kamhawi

co-Editor-in-Chief

Paul Brindley

co-Editor-in-Chief

Please update the inset map to include the country name. Additionally, add a latitude and longitude scale to the main map to clearly indicate the geographic location of the study area.

---

## [Editor Report · Acceptance letter]

Dear Dr. Sene,

We are delighted to inform you that your manuscript, "Entomological Surveys and Insecticide Resistance in the Dengue Vector Aedes aegypti in Dakar, Senegal: First Detection of the kdr Mutation," has been formally accepted for publication in PLOS Neglected Tropical Diseases.

Best regards,

Shaden Kamhawi

co-Editor-in-Chief

Paul Brindley

co-Editor-in-Chief
